# Estimating ocean currents from the joint reconstruction of absolute dynamic topography and sea surface temperature through deep learning algorithms

Daniele Ciani[1], Claudia Fanelli[2], and Bruno Buongiorno Nardelli[2]

[1]Consiglio Nazionale delle Ricerche, Istituto di Scienze Marine (CNR-ISMAR), 00133, Rome, Italy
[2]Consiglio Nazionale delle Ricerche, Istituto di Scienze Marine (CNR-ISMAR), 80133, Naples, Italy

**Correspondence:** Daniele Ciani (daniele.ciani@cnr.it)

**Abstract.** Our study focuses on Absolute Dynamic Topography (ADT) and Sea Surface Temperature (SST) mapping from satellite observations with the primary objective of improving the satellite-derived ADT (and derived geostrophic currents) spatial resolution. Retrieving consistent high resolution ADT and SST information from space is challenging, due to instrument limitations, sampling constraints and degradations introduced by the interpolation algorithms used to obtain gap free (L4) analyses. To address these issues, we developed and tested different deep learning methodologies, specifically Convolutional Neural Network (CNN) models that were originally proposed for single-image super-resolution. Building upon recent findings, we conduct an Observing System Simulation Experiments (OSSE) relying on Copernicus numerical model outputs (with respective temporal and spatial resolutions of 1 day and 1/24°) and we present a strategy for further refinements. Previous OSSEs combined low resolution L4 satellite equivalent ADTs with high resolution "perfectly known" SSTs to derive high resolution sea surface dynamical features. Here, we introduce realistic SST L4 processing errors and modify the network to concurrently predict high resolution SST and ADT from synthetic, satellite equivalent L4 products. This modification allows us to evaluate the potential enhancement in the ADT and SST mapping while integrating dynamical constraints through tailored, physics informed loss functions. The neural networks are thus trained using OSSE data and subsequently applied to the Copernicus Marine Service satellite derived ADTs and SSTs, allowing us to reconstruct super resolved ADTs and geostrophic currents at the same spatio-temporal resolution of the model outputs employed for the OSSE. A 12 years long time series of super resolved geostrophic currents (2008-2019) is thus presented and validated against in situ measured currents from drogued drifting buoys and via spectral analyses. This study suggests that CNNs are beneficial for improving standard Altimetry mapping: they generally sharpen the ADT gradients with consequent correction of the surface currents direction and intensities with respect to the altimeter derived products. Our investigation is focused on the Mediterranean Sea, a quite challenging region due to its small Rossby deformation radius (around 10 km).

## 1 Introduction

Oceanic currents play a pivotal role in influencing both short term and long term dynamics within the ocean-atmosphere system. Monitoring these currents on a large scale is essential for assessing the transport of heat and salt and enhancing our ability to

predict variations and shifts in ocean dynamics and their impact on marine ecosystem. At smaller oceanic scales, such as mesoscale and submesoscale, observing and predicting ocean currents is key to understand and model Earth system dynamics. Mesoscale eddies, oceanic features with length scales from 10 to 100 km and persisting on timescales from weeks to months, can migrate over considerable distances while carrying heat, salt, and nutrients. These eddies also introduce perturbations that drive significant vertical exchanges. Submesoscale features, including eddies, fronts, and filaments, operate at spatial and temporal scales of 0.1 to 10 km and hours to days. These transient, small-scale features introduce notable variations in both horizontal and vertical velocities, profoundly affecting local three-dimensional transport properties (Clarke and Li, 2004; Li and Clarke, 2004; Carlson and Clarke, 2009; Siokou-Frangou et al., 2010; Buongiorno Nardelli, 2013; Barbosa Aguiar et al., 2013; Ponte et al., 2013; Frenger et al., 2013; Bashmachnikov et al., 2015; Chenillat et al., 2016). The monitoring of ocean currents is also crucial for various societal and environmental purposes, including aiding ship navigation, supporting safety and rescue operations, as well as for the management of marine ecosystem services Pisano et al. (2016); Onink et al. (2019). These applications necessitate precise, high resolution tracking of surface oceanic currents.

Since the early 1990s, satellites equipped with nadir looking radar altimeters have provided indirect observations of the marine surface circulation at global scale. This is accomplished by measuring the Absolute Dynamic Topography (ADT) with respect to a reference surface (the geoid) along 1-D tracks and inferring surface motion from interpolated 2D ADT maps using the geostrophic approximation. However, this method has inherent limitations tied to ADT sampling and the geostrophic approximation, primarily capturing only larger mesoscale geostrophic processes, $O(100 \text{ km, 10 days})$ (Pascual et al. (2006); Pujol et al. (2012, 2016); Ballarotta et al. (2019b)).

The direct estimation of marine surface (or near surface) currents relies on satellite radar interferometry techniques or in situ measurements from Lagrangian buoys, ship mounted devices such as Acoustic Doppler Current Profiler (ADCP), or High Frequency Radar (HFR) platforms. Lagrangian observations can serve as reference points to validate remotely sensed surface currents and, when properly organized in space and time, can yield pseudo Eulerian surface circulation estimates. However, this approach faces constraints related to the spatial-temporal coverage of Lagrangian platforms and their tendency to become trapped in oceanic recirculation or convergence zones. Conversely, HFR systems offer comprehensive maps at fine spatial and temporal resolutions (less than 10 km and 1 hour, respectively), albeit only within coastal regions (Chapron et al., 2005; Falco and Zambianchi, 2011; Lumpkin et al., 2017; Laurindo et al., 2017; Capodici et al., 2019; Ribotti et al., 2023; Fanelli et al., 2024b).

The fusion of altimeter derived and in situ measured currents represents a valuable strategy for enhancing surface currents estimates from altimetry, in both coastal and open ocean regions (Mulet et al. (2021); Ballarotta et al. (2022)). Nonetheless, the effectiveness of this approach is contingent on the availability of in-situ measurements. In alternative, several studies proposed the merging of altimeter derived products with independently observed satellite derived tracers, like sea surface temperature (SST) and surface chlorophyll concentration data, (González-Haro and Isern-Fontanet, 2014; Rio and Santoleri, 2018; Ciani et al., 2020; González-Haro et al., 2020; Miracca-Lage et al., 2022). Such methodologies turned out to be useful in improving the altimeter derived geostrophic circulation with limitations related to the season and/or the geographic location, making it challenging to implement them operationally at global scale.

High resolution monitoring of ocean dynamic topography using imaging sensors, which natively provide two dimensional observations have only recently become available thanks to the successful launch of the Surface Water and Ocean Topography (SWOT) mission in December 2022 (https://swot.jpl.nasa.gov/mission/). This is currently providing 2D images of sea surface height at unprecedented spatial resolutions, allowing to better characterize the signatures of oceanic mesoscale features from altimetry maps. The SWOT revisit time is however set to 11 days, limiting the high rate monitoring of the fast evolving or persistent oceanic features (Fu et al., 2009; Fu and Ubelmann, 2014; Morrow et al., 2019, 2023).

Here, we propose an ocean surface currents reconstruction methodology based on Artificial Intelligence, also following recommendations from the international altimetry team (Abdalla et al., 2021) and recently implemented by Beauchamp et al. (2022); Buongiorno Nardelli et al. (2022); Martin et al. (2023); Fablet et al. (2023); Moschos et al. (2023); Kugusheva et al. (2024); Archambault et al. (2024); Martin et al. (2024).

In particular, we rely on a family of deep learning methods known as convolutional neural networks (CNN) for super resolution, firstly proposed for computer vision purposes (e.g., Dong et al., 2015). CNNs for super resolution learn a direct mapping between low resolution and high resolution images and are here applied to the case of satellite derived images of Absolute Dynamic Topography (ADT) and Sea Surface Temperature (SST). Our primary focus is to super resolve ADT maps exploiting information from low resolution interpolated ADTs and higher resolution interpolated SSTs, improving both in a joint reconstruction. Then, the super-resolved ADT maps are used to derive the oceanic surface circulation via the geostrophic balance. The study builds on, but also constitutes a substantial advance with respect to, the results firstly presented by Buongiorno Nardelli et al. (2022) (BBN22 hereinafter) which consisted in training a CNN through an Observing System Simulation Experiment (OSSE) and then using the same CNN model to reconstruct ocean surface currents from true satellite products. BBN22 also presented a sensitivity study to assess the performances of different CNN architectures, mainly testing networks with different numbers of tunable parameters and different perceptive capabilities. However, BBN22 performed the OSSE only considering the limitations of the altimeter products (i.e. accounting for the interpolation of observations in order to obtain 2D ADT maps from the along-track observations), thus assuming a perfectly known SST. Our work improves BBN22 on two particular aspects: i) we generate a satellite-equivalent SST time series accounting for a realistic operational level 4 (gap free) analysis based on gapped input data, as in Buongiorno Nardelli et al. (2013); ii) we train our CNN relying on updated loss-functions, which now include physics based constraints and exploit the joint reconstruction of ADT, SST and $\partial_t$SST data, also imposing physics-based constraints. The study focuses the Mediterranean Sea area, dominated by motions with length scales down to 6 km (Malanotte-Rizzoli, 2011), thus constituting a challenging test bed for the super resolution of standard Altimetry products.

## 2 Materials and Methods

The datasets involved in our study are basically those described in BBN22 and Ciani et al. (2021), with the exception of the newly generated synthetic SSTs described in section 2.5.2. For the sake of clarity, we provide a brief description of all datasets in the present manuscript as well.

## 2.1 Numerical Model

The Mediterranean Forecasting System (MFS) is a hydrodynamic model designed for the Mediterranean Basin and the easternmost section of the Atlantic Ocean near the Strait of Gibraltar. It provides 3D horizontal current and Sea Surface Height (SSH) outputs ranging from monthly to 15 minute intervals, as well as 3D temperature and salinity fields with monthly to hourly estimates. These data are accessible via the Copernicus Marine Service web portal (Product ID: MEDSEA-ANALYSIS-FORECAST-PHY-006-013). For our current research, we utilized daily SSH and SST data extracted within the boundaries of the Mediterranean Basin (30 to 46°N and 6°W to 37°E). These datasets are provided on a 1/24° regular grid with 125 unequally spaced vertical levels. The simulations are based on the NEMO model (Nucleus for European Modelling of the Ocean) in combination with Wave Watch-III for the wave component. The MFS simulations also incorporate data assimilation from 2D satellite derived SST, vertical salinity profiles, and sea level anomaly observations along satellite tracks, (Clementi et al., 2019).

## 2.2 Altimeter ADT and derived quantities

The Altimeter derived ADT were obtained from the Copernicus Marine data store. The surface currents, for consistency with the results presented in Section 3.2, were derived from ADT using the geostrophic approximation equation, i.e. by applying a finite central differences operator to the gridded L4 ADTs in order to compute its spatial gradients. The Copernicus ADTs are provided as daily data with a nominal 1/8° horizontal resolution. We extracted the time series covering the years 2008 to 2019. The corresponding Copernicus Marine Service product and dataset ID are SEALEVEL-MED-PHY-L4-REP-OBSERVATIONS-008-051 and dataset-duacs-rep-medsea-merged-allsat-phy-l4, respectively (accessed on 1[st] March 2021 and now included as part of the SEALEVEL-EUR-PHY-L4-MY-008-068 product). To match the resolution of the numerical model outputs used in our study, the Altimeter ADTs are up-sized to 1/24° via cubic interpolation.

## 2.3 Satellite SST

We obtained remotely sensed SST data from the Copernicus Marine Service (https://doi.org/10.48670/moi-00172, last accessed on 14 January 2022). These are L4 products, ensuring gap free estimates of the foundation temperature (i.e. at $\simeq 10$ m depth) provided on a regular grid, and are produced and distributed operationally in near real time. Specifically, we used 12 years of the ultra high spatial resolution (UHR) Mediterranean dataset spanning from 2008 to 2019, with a nominal resolution of 1/100° (Product ID: SST-MED-SST-L4-NRT-OBSERVATIONS-010-004-c-V2). This SST product is generated by combining night time images collected by satellite infrared sensors after rigorous quality control, removal of cloudy pixels and then applying an optimal interpolation algorithm. In order to homogenize the satellite SST L4 data with the ones used for training the neural network model, we evaluated the effective spatial scales captured by the model SST and filtered the UHR via a low pass, cut-off wavelength Lanczos filter to obtain comparable effective resolutions. Successively, the satellite SSTs were remapped onto the final (model) 1/24° grid through bilinear interpolation.

## 2.4 In-situ measurements

In situ measurements of sea surface currents were obtained from autonomous Lagrangian drifting buoys, which are transported
passively by ocean surface currents. During the buoy's drifting process, positional data are interpolated at regular intervals
(approximately every 30 minutes) using the kriging interpolation method developed by Poulain et al. (2012). Velocities are
subsequently computed through a finite differences method applied to the interpolated positions and are provided with six
hourly temporal resolution. The data covering the period of our study were originally provided by the Italian Institute of
Oceanography and Experimental Geophysics (OGS) for the ESA CIRCOL project (http://circol.artov.ismar.cnr.it/). These time
series are accessible via http://doi.org/10.6092/7a8499bc-c5ee-472c-b8b5-03523d1e73e9 (last accessed on 29 October 2023).
It is worth noting that buoy derived surface current values are retained only if the buoy is equipped with a drogue, a device that
ensures the buoy's movement is primarily driven by ocean currents rather than surface winds (Menna et al., 2018).

## 2.5 Synthetic Satellite Equivalent ADT and SST

### 2.5.1 Satellite equivalent ADT

We generated one year (2017) of synthetic, Satellite Equivalent altimeter derived Absolute Dynamic Topography (SE-ADT
hereinafter) maps using outputs from the Copernicus Marine Service MFS hydrodynamic simulation, employing the Data
Unification and Altimeter Combination System (DUACS) mapping method. The steps involved in this process are detailed
below. Initially, sea level anomaly (SLA) was calculated from model outputs according to the following expression:

$$SLA = SSH - (MDT - 0.344). \tag{1}$$

Here, the mean dynamic topography (MDT) is provided as a static field along with the model outputs. A constant value of
0.344 (expressed in m) is used to adjust the SLA values in the Mediterranean Sea to ensure that the spatio-temporal average
of SLA is zero for the year 2017. To remove the large scale, high frequency variability typically due to dynamic atmospheric
processes, we applied a Loess filter to these synthetic data. Subsequently, the SLA was sampled along the actual paths of a
synthetic constellation comprising four radar altimeters: Jason-3, Sentinel-3A, SARAL/Altika, and Cryosat-2 missions. This
step was executed using the SWOT simulator software, which accounts for the actual orbits, errors, and noise associated
with each mission. The chosen four satellite constellation represents the constellation used in the Copernicus Marine Service
processing during 2017. These along track synthetic measurements were then incorporated into the DUACS processing chain
to generate L4 SLA maps. The optimal interpolation (OI) scheme follows the DUACS DT2018 (Delayed Time) configuration
for the Mediterranean area, as described in Taburet et al. (2019). The reconstructed L4 maps were then combined with the
filtered large scale maps in order to obtain the ADT. These data are provided on a daily basis and are available on a regular
1/8° grid (further details can be found in Ciani et al. (2021)). A subsequent up-size to 1/24° is applied before using this dataset
for our application.

### 2.5.2 Satellite equivalent SST

The generation of the synthetic satellite equivalent SST (SE-SST) follows the same principle as for the SE-ADT. The SE-SSTs combine information from the model derived SST and the Copernicus Marine Service merged multi sensor L3S (Level 3, Super-collated) satellite SSTs for the Mediterranean Area (product ID: SST_MED_SST_L3S_NRT_OBSERVATIONS_010_012, accessed on 20 October 2023). The L3S SSTs are a satellite derived product that consider information from several infrared radiometers and provide daily ocean surface temperature on a regular latitude-longitude grid at high (1/16°) and ultra high (1/100°) spatial resolution, representative of nighttime SST values (00:00 UTC). As such, the L3S SST contains gaps whenever the infrared SST retrieval is not possible (e.g. due to cloud cover) or the single sensor satellite SSTs have been labeled as poor quality observations. Starting from one year of model derived SSTs, we remapped gappy (missing values) patches found in the 1/16° L3S product (year 2017) onto the original modelled SST, thus generating a synthetic model derived L3S SST time series. Finally, the gap-free (Level 4) SE-SSTs, including an estimate of the uncertainty on the Level 4 analysis, are obtained through standard OI via a dedicated algorithm reproducing the main steps described in Buongiorno Nardelli et al. (2013), constituting the backbone of the present day L4 satellite SST operational production for the Mediterranean Area. An inter comparison of the synthetic gapped SST, SE-SST generated via OI and ground truth SST is sketched in Fig. 1, where we notice a strong smoothing and underestimation of the mesoscale SST features in correspondence of cloudy areas, as expected for present L4 SST satellite products. In the end, the $\partial_t$(SE-SST) and its error are computed from the SE-SST time series respectively via central finite differences and standard error propagation. The SE-SST, $\partial_t$(SE-SST) and their errors are daily fields provided on 1/24° regular grid.

### 2.6 Ocean currents and SST reconstruction methodology

In this study, ocean currents are reconstructed through a joint ADT/SST super resolution relying on a CNN approach, with a primary focus on ADT. The workflow of the reconstruction exercise is sketched in Fig. 2. The work consists in three phases: i) the satellite equivalent data generation (detailed in Sections 2.5.1 and 2.5.2) ; ii) the training of the network by means of an Observing System Simulation Experiment (OSSE); iii) the application of the neural network model (optimized via the OSSE) for the reconstruction of the ocean surface currents from satellite derived input ADTs and SSTs. The OSSE consists in the generation of super resolved ADTs/SSTs which are then directly compared with an independent test dataset detailed in section 2.6.1. In addition, in the present study we will also inter-compare our results with the ones obtained by BBN22, i.e. an earlier version of the dilated adapative multi-scale residual Super Resolution CNN, detailed in Section 2.6.1. The Super-Resolved geostrophic currents obtained from satellite derived data are provided at 1/24° (to match the data used for training the neural network with the OSSE) and their performance assessment is carried out via direct comparison with in situ measured currents (see Section 2.4) and spectral analysis.

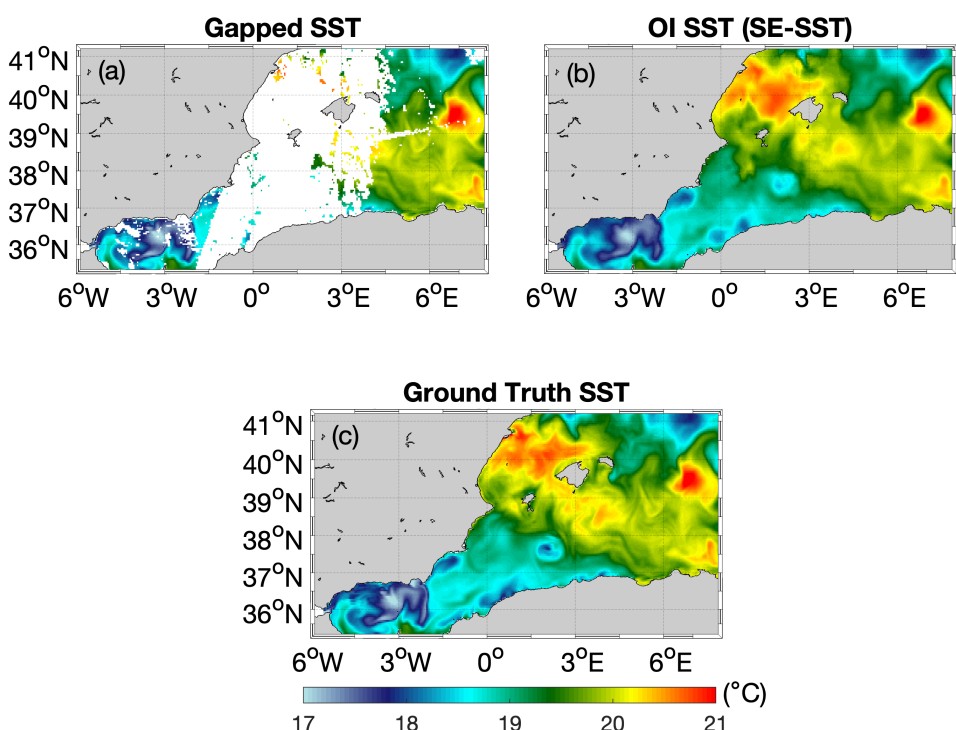

**Figure 1.** Results of the OSSE for the Satellite Equivalent (SE) SST generation on 16 May 2017. (a): SE L3S gapped SST; (b): SE-SST generated via OI algorithm; (c): ground truth SST.

### 2.6.1 Convolutional Neural Network Architecture and training strategy

Convolutional Neural Networks used for super resolution are built to learn a mapping between low resolution and high resolution images. This is achieved relying on convolution operators designed to detect and learn specific features in the input images. The network learns the end to end mapping from low to high resolution images via a minimization of the error between the output and the validation data. This is quantitatively achieved using Loss functions which are iteratively minimized relying on a validation dataset (i.e. a fraction of the dataset involved in the training procedure). The overall performance of the CNN

is finally quantified by means of an independent test dataset.

The dataset preparation is carried out as follows: i) firstly, we extracted 40 dates from the model outputs detailed in Section 2.1, which served as independent test dataset. The 40 dates are distributed along the year in order to cover all dynamical regimes; ii) starting from the remaining model data, we performed a resampling into $76 \times 100$ pixels (corresponding to $\simeq$ $300 \times 400$ km) tiles. The tiles extraction also considered a 50% spatial overlap, resulting in a total of 42250 samples. All

195 the samples then undergo a normalization procedure with respect to the maximum observed in the time series. In addition, for SE-ADT, SE-SST and $\partial_t$(SE-SST), we compute anomalies filtering out signals larger than approximately 200 km. This

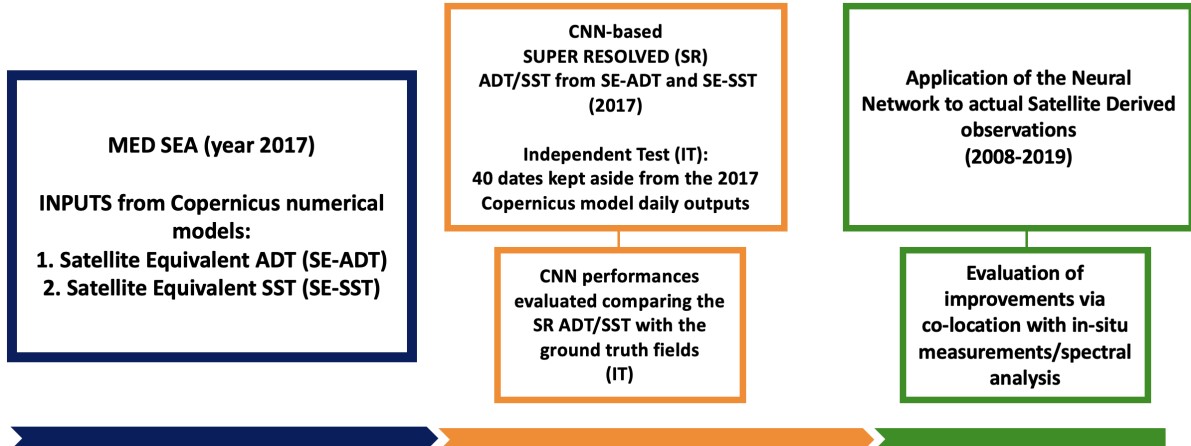

**Figure 2.** Workflow of the study. Blue stands for generation of the satellite equivalent input data; orange blocks indicate the OSSE and green blocks refer to the application of the reconstruction methodology to satellite derived data.

operation aims at retaining signals related to the local SST variations due to the horizontal currents advection. In this way the extraction of patterns from the SST field can help the ADT reconstruction with contributions primarily due to the ocean surface dynamics. Such samples were thus used for network training/validation procedure. Here, compared to BBN22, we changed the splitting strategy between training and validation datasets, whose relative amount is respectively 85% and 15% of the 42250 (total) samples. In particular, we forced the validation dataset to be a time series of samples adjacent in time (during the late fall/early winter season), instead of applying a random selection from the available samples. This strategy is used to train the network preferably selecting scenarios where enhanced small-mesoscale/ submesoscale activity is expected, as further detailed in Section 4.

The CNN employed here is called dilated Adaptive Deep Residual Network for Super-Resolution (dADR-SR), sketched in Fig. 3. In previous formulations (BBN22), the network considered:

- four predictors: namely the SE-ADT, SE-ADT error, SST and its temporal derivatives ($\partial_t SST$);

- one target, i.e. the super-resolved ADT.

The predictors were chosen considering the (local) modulation of SST features by water mass advection as well as the present-day altimeter observation geometry and satellite observations repetitiveness. The main upgrade of the present study is the introduction of additional inputs (predictors) like the SE-SST and $\partial_t$(SE-SST) errors as well as two additional target-s/outputs, i.e. the super resolved SST and its temporal derivatives ($\partial_t$(SE-SST)). This upgraded architecture, sketched in Fig. 3, allows to perform a more realistic ADT/SST reconstruction exercise, also accounting for the limitations of the present day satellite SST operational retrieval.

The basic architecture of the CNN is briefly recalled here. In the dADR-SR network, the low resolution input dataset initially passes through three parallel dilated convolutional layers, each containing 10 convolutional filters characterized by a 3×3 kernel and an increasing dilation factor (1, 3, and 5, respectively). Dilated convolution enhances the CNN perceptive capability, enabling the extraction of information at increasingly larger scales maintaining the same computational cost of a standard 3×3 convolutional filter. Following this initial stage, the data undergo a sequence of twelve Multiscale Adaptive Residual Blocks (MARB). Each of these blocks incorporates two sets of parallel dilated convolutional layers, featuring 120 and 10 filters, respectively, also including a Squeeze-and-Excitation (SE) module. The SE block functions as a channel attention mechanism, adaptively scaling information coming from the several input data before summing all contributions to produce the final high-resolution output.

The algorithm is trained applying an early stopping rule, terminating the training process when the validation loss function increases for a predefined number of epochs, as determined by the patience parameter, set to 5 in our study. An adaptive learning rate (initialized at lr = $10^{-4}$) is employed, and we rely on the Adam optimizer with the same configuration used in BBN22. The dADR-SR training model ultimately utilizes nearly 1.6 million trainable parameters.

We adopt mean squared error as initial loss function (LF), although physics informed constraints are introduced, as detailed in Section 3.

Finally, for the CNN test/prediction phase, we perform a tile by tile reconstruction and we merge the output high resolution tiles (considering a weighted average in the overlaps) in order to produce maps covering the entire study area. This is done on a tile-by-tile basis, with each tile incorporating a 50% overlap in both the longitudinal and latitudinal directions. During this process, the central regions of the tiles are expected to achieve superior performance, whereas edge effects can lead to spurious features near the tile boundaries related to the application of convolutional kernels. To mitigate these edge-related artifacts, a pixel-wise weighting function is applied, progressively decreasing the weight assigned to pixels with increasing distance from the tile centers. This approach enables a seamless basin-wide reconstruction from the 76×100 tiles.

## 3 Results

### 3.1 The Observing System Simulation Experiment

The results of the OSSE are presented below. We firstly discuss a test case for the ADT reconstruction on 4 January 2017, skecthed in Figs. 4 and 5.

The ADT maps obtained through CNN for super resolution (panel (b)), compared with the ones given by standard altimetry processing (in panel (a)) exhibit features in good agreement with the model outputs (our ground truth, panel (c)). Visual inspection suggests an overall sharpening of the basin scale ADT gradients and the potential of CNN approach to overcome the dynamical feature distortion due to standard altimetry mapping. This is observed in many cases along the Algerian coast, in proximity of the Gulf of Lion, as well as in the Adriatic Sea and the Levantine basin. Choosing a land free area in the Levantine Basin (sketched in Fig. 4(c)), relying on Fast Fourier Transform (FFT) analysis, we also quantified the spectral properties of the three aforementioned ADT estimates, as in Droghei et al. (2018). At large scale ($< 200 \, \mathrm{km}$) the three spectra

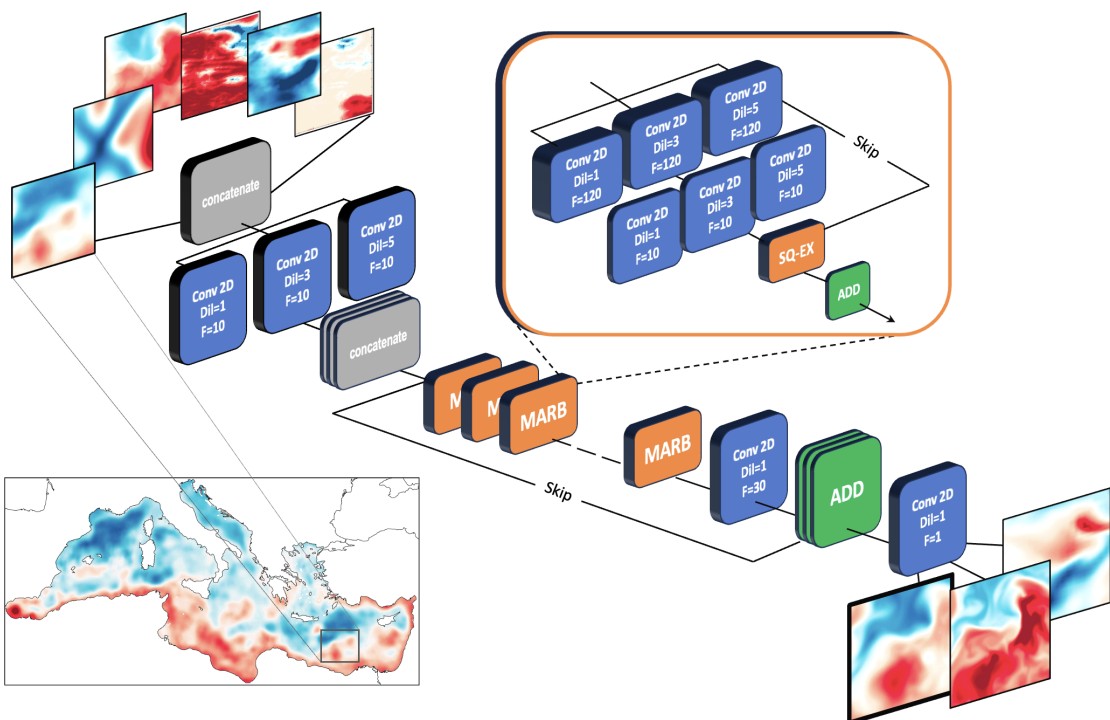

**Figure 3.** CNN architecture. The set of input tiles, from left to right, respectively indicate: SE-ADT, SE-ADT error, SE-SST, SE-SST error, $\partial_t$(SE-SST) and the $\partial_t$(SE-SST) error. The three output tiles, from left to right, respectively indicate the SR-ADT, SR-SST and SR-$\partial_t$SST. In the figure: Conv2D stands for 2D convolutional filter; Dil stands for dilation factor; F is the number of filters; ADD stands for aggregation; SQ-EX is the squeeze and excitation module and MARB indicates a multiscale adaptive residual block (further detailed in the panel highlighted by the dashed lines). The ADT output file is emphasized by a thick edge to indicate the focus of the present study.

exhibit a similar power spectral density (PSD), indicating a similar description of the largest mesoscale oceanic features. Progressively approaching smaller scales, i.e. from $\simeq 100$ km downward (1 $\deg^{-1}$ wavenumber onward), the Super Resolved

ADT spectrum (SR-ADT, red line in Fig. 4 (c)) evolves in fair good agreement with the ground-truth (green line in Fig. 4 (c)), confirming an improved representation of smaller mesoscale features compared to standard altimetry products. The SR-ADT spectrum eventually shows the injection of noise below scales of $\simeq 20$ km, as confirmed by a flattening of the spectrum. The aforementioned sharpening of the super resolved ADT gradients is further confirmed by the analyses reported in Fig. 5. In particular, we show maps of geostrophic currents velocity derived from the SE-ADT, the Super Resolved ADT and the

ground truth ADT, respectively shown in panels (a)-(c). The SE geostrophic current velocity is characterized by flows seldom exceeding $0.75 \, \mathrm{m\,s^{-1}}$ with broader spatial patterns compared to the ones derived from the ground truth ADT. On the other hand, the CNN super-resolved currents velocity (Fig. 5 (b)) exhibit sharper patterns at the basin scale, with flow intensities reaching 1 $\mathrm{m\,s^{-1}}$, in agreement with our reference field, given in Fig. 5 (c). Accordingly, performing the spectral analysis on the currents velocity fields (Fig. 5 (d)) leads to the same results already discussed for ADT. The spectral analyses presented in Fig. 4 and

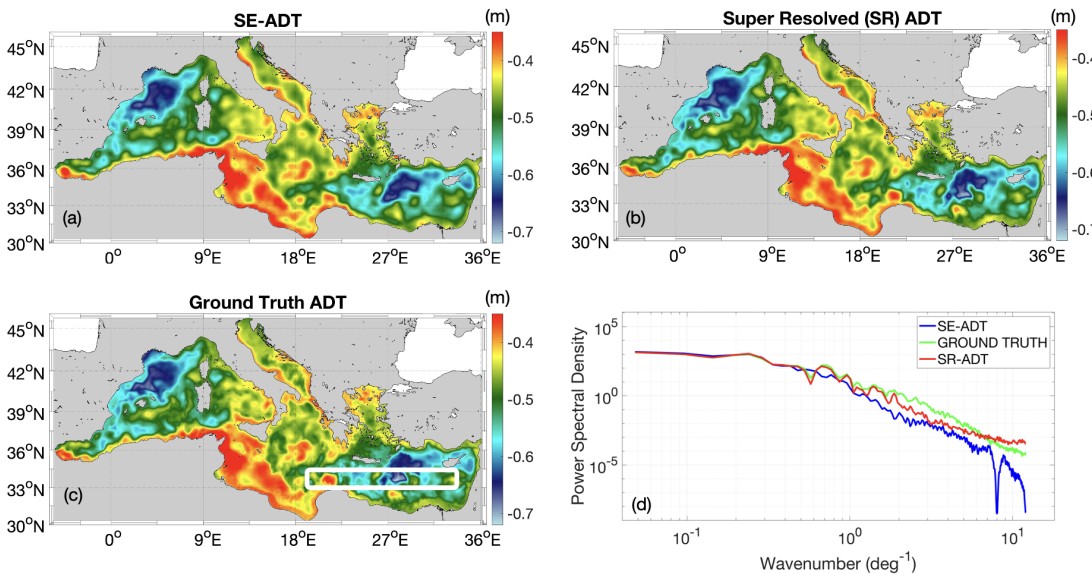

**Figure 4.** Results of the OSSE on 4 January 2017. (a): SE-ADT; (b): Super Resolved ADT; (c): model ADT (ground truth); (d): comparative spectral analysis of the ADT maps (SE-ADT, SR-ADT and ground truth are given in blue, red and green, respectively). Results refer to the 2D box depicted in panel (c).

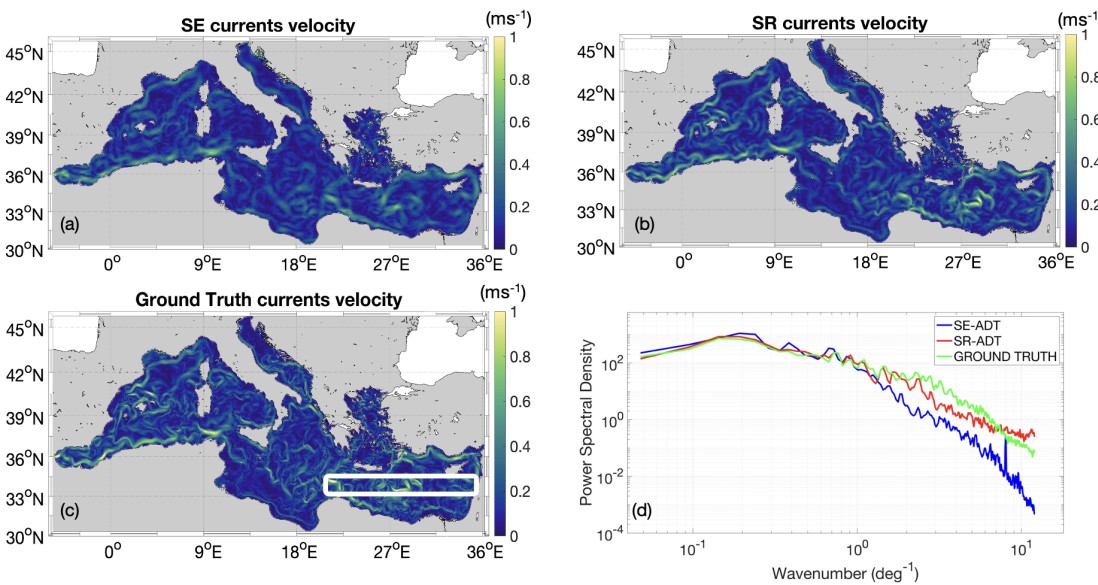

**Figure 5.** Results of the OSSE on 4 January 2017. (a): Satellite - Equivalent (SE) currents velocity from SE-ADT; (b): currents velocity from Super-Resolved (SR) ADT; (c): currents velocity from model ADT (ground truth); (d): comparative spectral analysis of the currents velocity maps (SE-ADT, SR-ADT and ground truth are given in blue, red and green, respectively). Results refer to the 2D box depicted in panel (c).

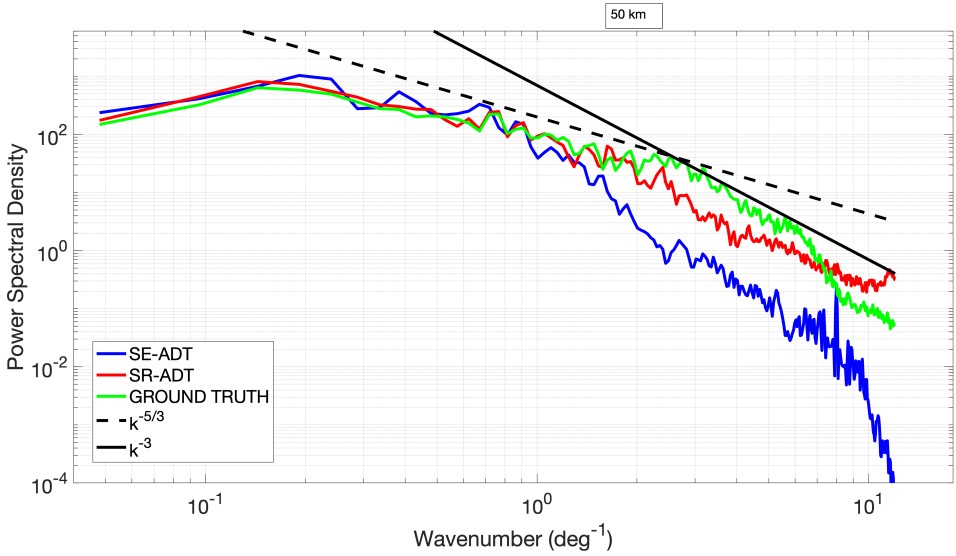

**Figure 6.** Results of the OSSE on 4 January 2017. Kinetic Energy spectra of the surface currents derived from: Satellite Equivalent (SE) ADT (blue); Super Resolved (SR) ADT (red); model outputs, ground truth ADT (green). Results refer to the 2D box depicted in figure 4(c). The continuous and dashed black lines refer to the $k^{-3}$ and $k^{-5/3}$ slopes, respectively. The 50 km scale is reported for reference.

5 serve to assess the enhancement of the mesoscale activity in the SR fields compared to present-day altimetry mapping. A more in-depth comparison with theoretical predictions of energy and enstrophy transfer is based on the computation of Kinetic Energy (KE), sketched in Fig. 6. We firstly compute the KE relying on the surface currents from the SE ADT, SR ADT and from the model outputs (ground truth). The KE spectra are then presented along with the $k^{-5/3}$ and $k^{-3}$ slopes, as in Ciani et al. (2019) and following Vallis (2006). With similar outcomes with respect to previous analyses, the ground truth KE (green

line in Fig.6) is in agreement with the predictions of energy/enstrophy transfer, following the $k^{-3}$ slope for larger wavenumbers and the $k^{-5/3}$ for shorter ones. An expected exception occurs for wavenumbers approaching $10 \ \mathrm{deg}^{-1}$, where the spectrum exhibits an energy loss, suggesting that model outputs are unable to fully resolve the small mesoscale motion. The KE spectra derived from SR and SE ADTs, are superimposed to the ground truth case only in the large mesoscale regime (wavenumbers $< 1 \ \mathrm{deg}^{-1}$). As soon as the mesoscale regime is reached, the SE KE spectrum experiences a significant loss of energy, while

the SR KE follows theoretical predictions until scales of $\simeq 50 \ \mathrm{km}$, which is thus identified as the scale at which processes are fully resolved.

     In addition, we present the CNN performances as Root Mean Square Error (RMSE) differences, $\Delta$RMSE, between two versions of the ADT L4 mapping, using the independent test data (i.e. the original model derived ADTs) as benchmark. The aforementioned L4 ADT mappings are: i) the operational standard altimetry system discussed in section 2.5.1 and ii) the CNN

based reconstruction proposed in the present study. In practice, whenever the $\Delta$RMSE assumes positive values, the CNN-based reconstruction improves the capabilities of the standard altimetry system. In particular, Fig. 7 illustrates a comparative analysis

between the optimal CNN reconstruction proposed by BBN22 (top panel) and the one resulting from upgrading the CNN architecture (bottom panel).

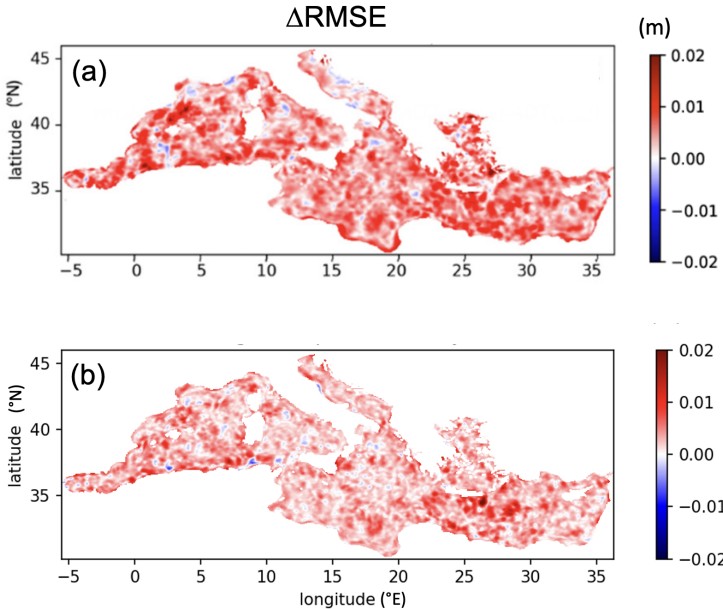

**Figure 7.** Top: $\Delta$RMSE (in m) according to the CNN architecture of BBN22. Bottom: $\Delta$RMSE (in m) following the CNN proposed in the present study. Positive values indicate an ADT mapping improvement with respect to standard Altimetry.

In the BBN22 formulation, the OSSE indicates an overall improvement of the CNN-based reconstruction, with very few spots of degradation mainly distributed along the coasts of the Adriatic Sea, the Liguro-Provencal area and few additional isolated spots distributed in the southern section of the basin (from 30 to 35 °N). Upgrading the CNN still enabled to observe improvements in more than 95% of the basin, with isolated spots of degradation found in the western Mediterranean sea and an overall decrease of the more intense $\Delta$RMSE values. However, $\Delta$RMSE areas around 2 cm, expressing an improvement of 60% with respect to standard altimetry (computed as in Rio and Santoleri (2018)) are ubiquitous. The slight decrease of the $\Delta$RMSEs compared to the BBN22 formulation is not surprising, as we presently train the neural network based on a degraded, satellite equivalent SST field, which likely impacts on the overall CNN performances with the advantage of including information on the actual, current SST observing systems, also including a new training strategy which can likely impact the results at the OSSE level, as described thoroughly at the end of this section. Upgrading the CNN architecture, beyond relying on a realistic observing system, also enables the fine tuning of the LF during training. In the present day CNN formulation, the LF is given by the following equation:

$$\text{LF} = \alpha\left[\overline{(\text{SST}_{\text{pred}} - \text{SST}_{\text{ref}})^2}\right] + \beta\left[\overline{(\text{ADT}_{\text{pred}} - \text{ADT}_{\text{ref}})^2}\right] + \gamma\left\{\overline{[(\partial_t\text{SST})_{\text{pred}} - (\partial_t\text{SST})_{\text{ref}}]^2}\right\} + \delta\text{Loss}_{\text{phy}} \quad (2)$$

where:

- $\text{Loss}_{\text{Phy}} = \overline{\left( \left( \frac{\partial \text{SST}}{\partial t} \right)_{\text{pred}} - \frac{g}{f} \frac{\partial \text{ADT}_{\text{pred}}}{\partial y} \frac{\partial \text{SST}_{\text{pred}}}{\partial x} + \frac{g}{f} \frac{\partial \text{ADT}_{\text{pred}}}{\partial x} \frac{\partial \text{SST}_{\text{pred}}}{\partial y} \right)^2}$ ;

- $\alpha$=1, $\beta$=0.25, $\gamma$=0.67, $\delta$=0.025 ;

- the subscripts "pred" and "ref" respectively stand for the output of the CNN prediction and the validation dataset used during training;

- g,f,x,y,t are respectively the gravity acceleration, the Coriolis parameter, the zonal, meridional and temporal coordinates, respectively.

As such, the LF employed in our study quantifies the discrepancies between the predicted and the ground truth ADTs, SSTs, $\partial_t$SSTs, and includes an additional physics informed term ($\text{Loss}_{\text{phy}}$ in Equation (2)). Locally, this is equivalent to imposing the evolution of the ocean surface temperature according to the horizontal geostrophic advection on the super resolved fields predicted by the CNN, based on the following equation:

$$\frac{\partial \text{SST}}{\partial t} + u_g \frac{\partial \text{SST}}{\partial x} + v_g \frac{\partial \text{SST}}{\partial y} = 0, \tag{3}$$

where:

- $(u_g, v_g)$ indicate the zonal and meridional surface geostrophic currents, respectively;

- x,y,t stand for zonal, meridional and temporal coordinates, respectively.

The physics informed LF is applied to the super resolved fields predicted by the CNN. The coefficients $\alpha$, $\beta$, $\gamma$, $\delta$ have been determined via preliminary CNN training over 3 epochs, considering separately the four terms of the LF appearing in (2). In particular, we firstly impose $\alpha$=1 and perform a first training of the CNN accounting only for SST. We then repeat the same exercise for the remaining terms of the LF separately, and we estimate the mean ratio of the training and validation losses compared to the SST case. We finally assign those ratios to the coefficients $\beta$, $\gamma$ and $\delta$. The aim of this operation is to allow the CNN to be trained considering the weighted contributions from the four terms appearing in the LF (more details on the LF optimization are provided in Appendix A).

As mentioned in section 2.6.1, in the present study we modified the CNN training strategy systematically excluding 15% of tiles towards the end of our available time series (year 2017). This is primarily due to the potential over-fitting issue recently claimed by Martin et al. (2023). In order to quantify the CNN performances throughout the year, we computed daily, basin-scale percentages of improvement (again, as in Ciani et al. (2021) and Rio and Santoleri (2018)) to quantify the relative improvement of the CNN reconstruction with respect to standard altimetry along the 40 dates of our independent test dataset, as reported by Fig. 8.

As expected, the percentage of improvement drops after Julian day 300, in correspondence of the time indices for which tiles had been excluded during training/validation. However, the improvement is still exhibiting satisfying basin scale improvements,

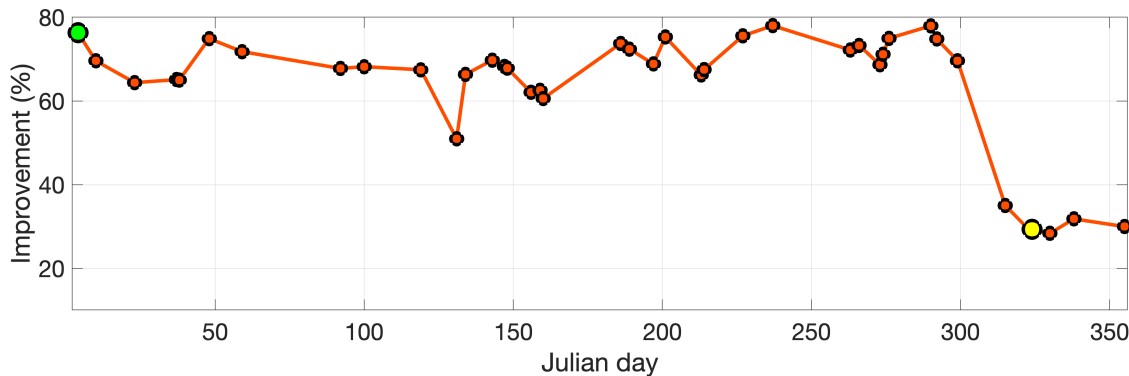

**Figure 8.** Basin-scale percentage of improvement of the CNN reconstruction with respect to standard altimetry. The improvement is evaluated against the independent test dataset. The green and yellow dots respectively stand for julian day 4 and 324, and refer to the case studies respectively depicted by Figs. 5 and 9.

around 30%, indicating that the CNN prediction is still efficient in reconstructing features unseen during the training phase. Performing the spectral analysis, as depicted in Fig. 5, for Julian day 324 (20 November 2017), revealed differences in the CNN performances compared to the temporal window bounded by Julian days 1 and 310. In general, the SE-ADT spectrum

is closer to the ground truth characteristics over the entire wavenumbers range. However, within the range of 1 to 4 $\mathrm{deg}^{-1}$ (approximately 100 to 30 $\mathrm{km}$), the SR-ADT spectrum demonstrates better agreement with the ground truth in comparison to the satellite equivalent spectrum. This reaffirms the CNN ability to enhance the characterization of mesoscale dynamics (Fig. 9 (d)). The potential overfitting issue will be further addressed in future studies, in which we plan to extend the time series of the OSSE. Ideally, we aim at exploiting one full year for each of the following operations: training, validation and test.

### 3.1.1 Reconstruction of ocean surface temperature

Although our primary scope is the computation of super-resolved surface currents from super resolved ADTs, we briefly illustrate the CNN performances in reconstructing the super resolved SSTs, which is among the novelties of the new CNN architecture. This is summarized by Fig. 10.

Overall, the CNN is indicating an improvement through the Mediterranean Basin, as illustrated by the dominance of positive

$\Delta$RMSE values. The local $\Delta$RMSE maxima reached 0.1°C, corresponding to $\simeq 60\%$ improvement with respect to standard optimal interpolation mapping, while the mean basin-scale improvement is $\simeq 14\%$ . Unlike ADT, the appearance of local degradation areas looks ubiquitous. These small-scale degradation spots are likely due to the assumptions behind our study. As claimed in section 2.6.1, we build the CNN architecture considering the modulation of SST features by the surface geostrophic advection and, simultaneously, feeding the SST mapping with the dynamical information contained in ADT. This could limit

the description of SST signals related to non-geostrophic phenomena or to vertical advection. Such behaviour is under investigation and will be matter of future studies.

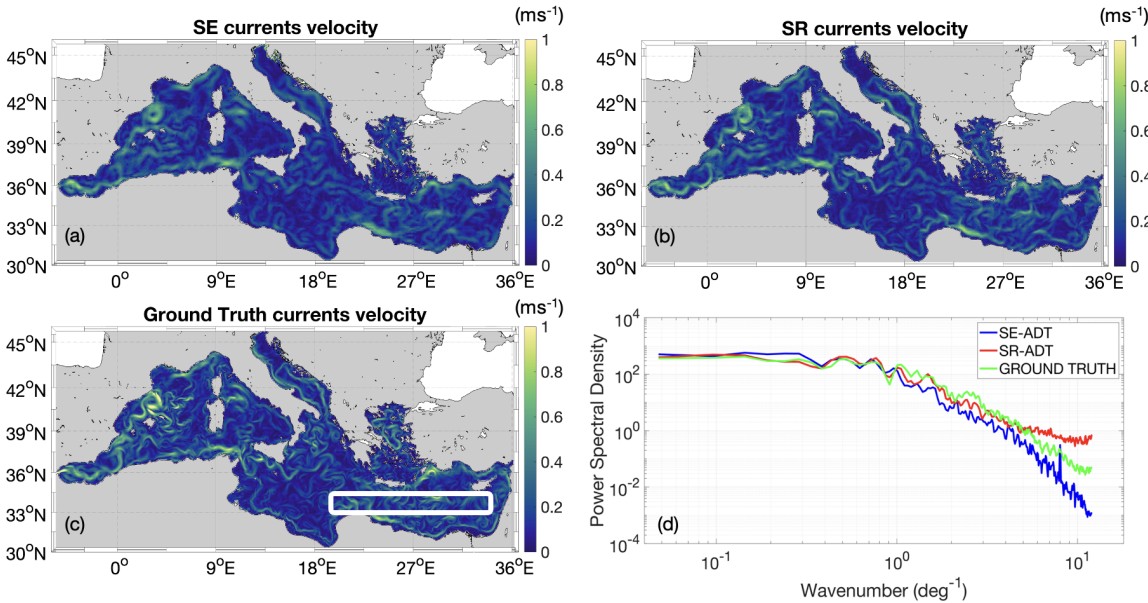

**Figure 9.** Results of the OSSE on 20 November 2017. (a): Satellite - Equivalent (SE) currents velocity from SE-ADT; (b): currents velocity from Super-Resolved ADT; (c): currents velocity from model ADT (ground truth); (d): comparative spectral analysis of the currents velocity maps. Results refer to the 2D box depicted in panel (c).

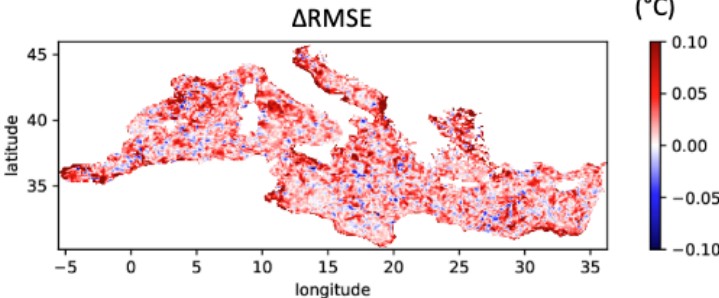

**Figure 10.** ΔRMSE (in °C) based on the CNN proposed in the present study. Positive values indicate an SST estimate improvement with respect to standard optimal interpolation.

## 3.2 Surface currents predictions from satellite derived data

The neural network model trained via the OSSE is now tested to predict super resolved ADTs, using state of the art L4 ADT derived from satellite altimetry and high resolution L4 satellite SSTs. The satellite derived input data are produced within the Copernicus Marine Service and cover the 2008 to 2019 timeframe (as detailed in Sections 2.2 and 2.3). The evaluation of CNN performance involves deriving super-resolved (SR) geostrophic currents from SR ADTs, through the geostrophic approximation. This is achieved applying a finite central differences operator to the SR ADTs (to estimate its spatial gradients) and using drifting buoys measurements as validation benchmark (detailed in Section 2.4). The validation relies on root mean square error inter comparisons, carried out interpolating the gridded altimeter derived and SR currents along the six hourly drifter acquisitions (for both components of the circulation) at the drifter time and location (as in Rio and Santoleri (2018)). In order to maximise the number of match ups between gridded and in-situ measured currents, statistics are provided in $2° \times 2°$ boxes, choosing a 12 years long timeseries (2008-2019). This guarantees a coverage of in situ measurements as depicted in Fig. 11, with an approximate minimum number of 100 observations per box.

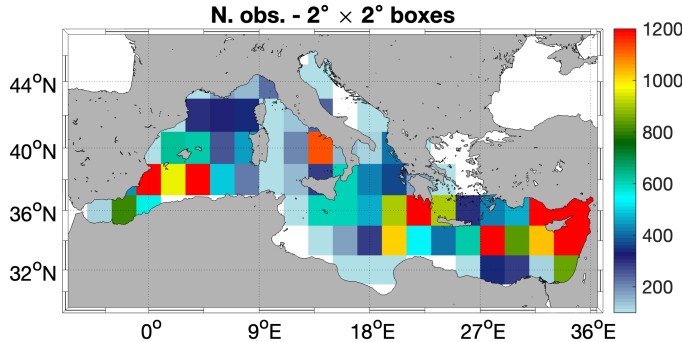

**Figure 11.** Number of in-situ measurements from drifting buoys in the 2008-2019 timeframe.

As addressed by BBN22, provided clear sky conditions, the overall effect of the CNN for super resolution is a sharpening of the mesoscale SR-ADT gradients compared to standard altimetry processing, with a significant enhancement of the mesoscale eddy activity in the derived SR currents. For clarity, a visual comparison of the Altimeter-derived currents (i.e. derived from the ADT computed via standard mapping) and SR currents velocity is provided in Fig. 12, and refers to 19 May 2019. Among the features of interest, we notice a significant intensification of the circulation north of the Balearic Islands, along the Algerian Currents and in the Sicily Strait, as well as a sharpening of the mesoscale features populating the southern Tyrrhenian Sea.

The SR currents, compared to standard altimeter-derived mapping, reduce the RMS by about 2 to 8 $\mathrm{cm\ s^{-1}}$ in the majority of the basin for both the zonal and meridional flow (Fig. 13). Degradations with respect to the altimeter (Alti) currents mainly occur in coastal areas and are particularly pronounced in the eastern tip of the Levantine basin. The basin scale RMS errors of the Alti and SR currents, reported in Table 1, evidence an improvement of the SR currents presented here, with basin scale reduction of the RMS error up to $\simeq 1\ \mathrm{cm\ s^{-1}}$.

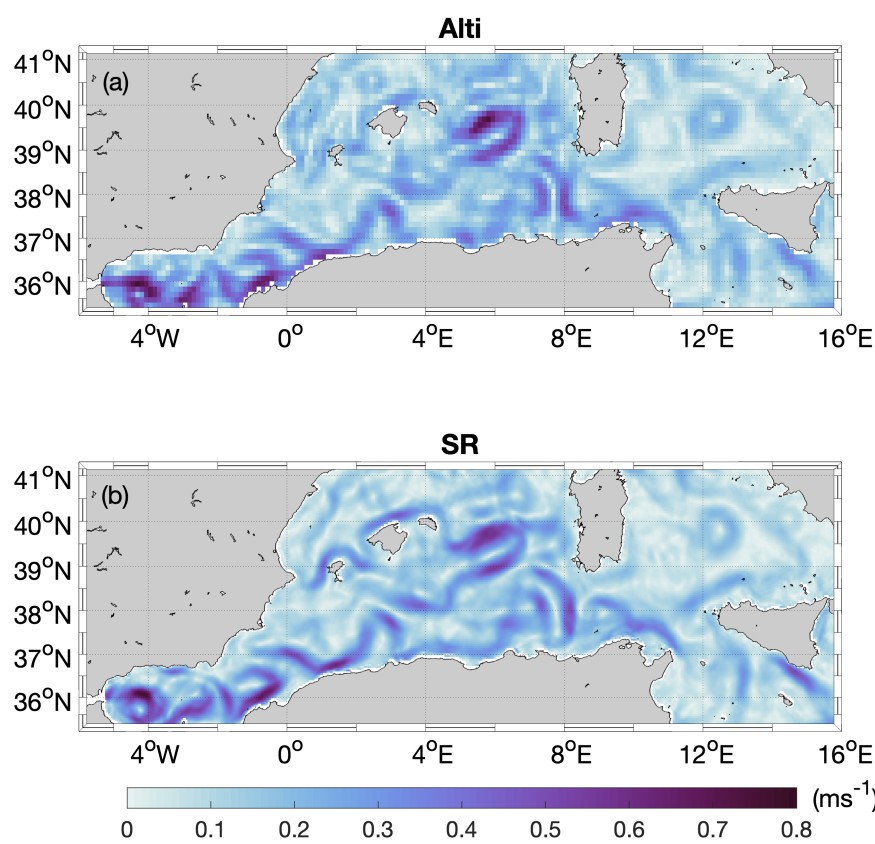

**Figure 12.** (a) Altimeter derived surface currents velocity; (b) SR currents velocity. The fields refer to 19 May 2019.

**Table 1.** RMS of the Altimeter derived (Alti) and Super Resolved (SR) surface currents, computed in $2° \times 2°$ boxes against in-situ measured currents. SR-BBN22 and SR respectively refer to the CNN ADT mapping described in Buongiorno Nardelli et al. (2022) and the one employed in the present study. U and V stand for zonal and meridional currents, respectively.

| MAPPING | Alti | SR-BBN22 | SR |
|---|---|---|---|
| **RMS U** (cm s$^{-1}$) | 12.34 | 12.30 | 11.90 |
| **RMS V** (cm s$^{-1}$) | 12.81 | 12.80 | 11.85 |

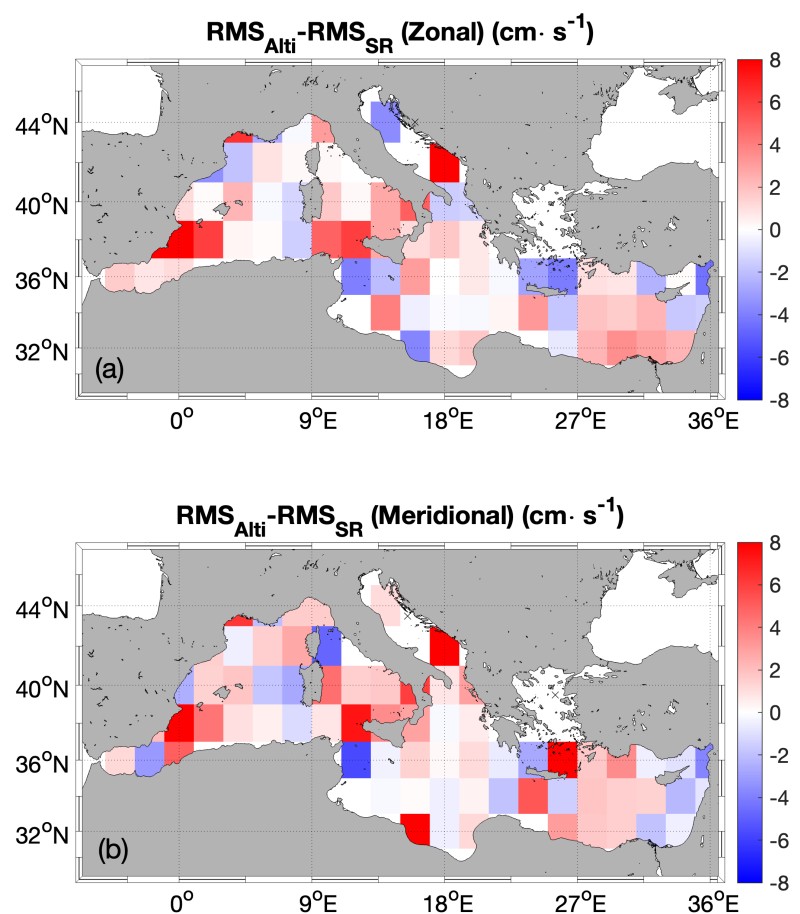

**Figure 13.** Differences of RMS errors between the Altimeter derived (Alti) and SR currents: (a) zonal flow, (b) meridional flow. Red areas express an improvement with respect to standard altimetry.

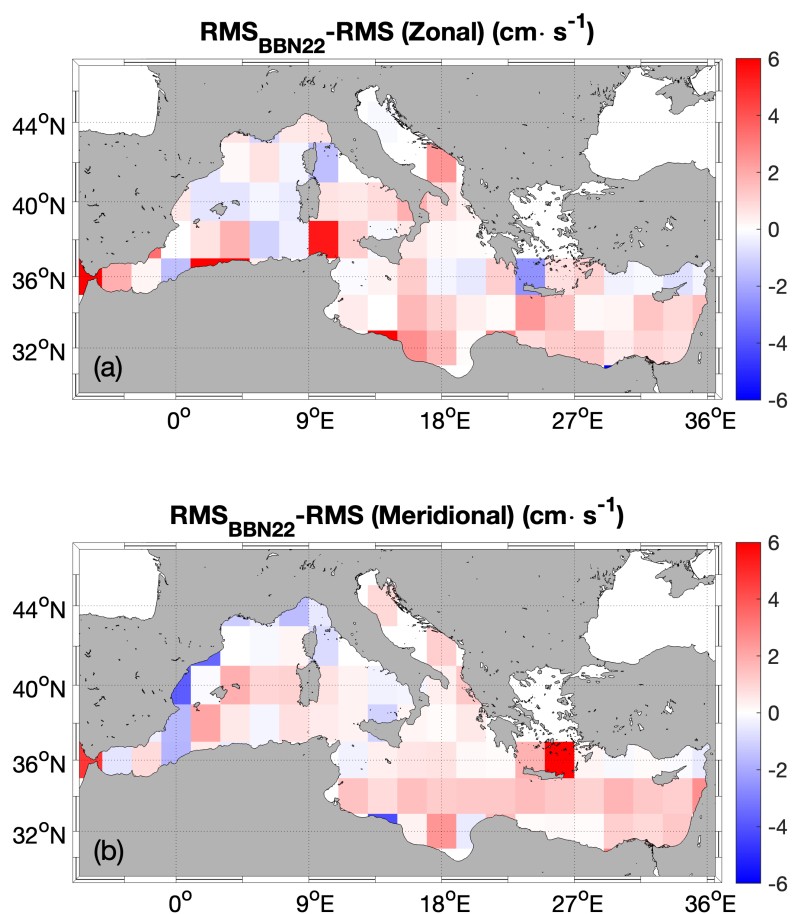

**Figure 14.** Differences of RMS errors between the BBN22 and the current CNN architecture: (a) zonal flow, (b) merdional flow. Red areas express an improvement of the current CNN compared to BBN22.

Improvements with respect to the BBN22 results are also observed. This is expressed by the overall RMS reduction of the new SR currents reported in Table 1 and is supported by the spatial distribution sketched in Fig. 14, in which red areas indicate an improvement of the current CNN formulation compared to BBN22. Such improvements cover 70% of our study area for the two components of the flow, and suggest an overall enhancement of the CNN performances mainly in the central and eastern Mediterranean, with occurrences of weak degradations ($\leq 1$ cms$^{-1}$) limited to the Western basin. Local occurrences of significant performances reduction with respect to BBN22 can be observed in coastal areas and indicate a worsening of the present-day SR currents RMS by about 5 cm s$^{-1}$ for the meridional flow. Nonetheless, an overall basin scale improvement is found.

Specific spectral analyses were carried out to inter-compare the surface kinetic energies (KE) derived from standard Altimeter (up-sized to the 1/24° grid) and super-resolved ADTs, to have insights on the effective spatial resolution of the two datasets. This analysis was performed estimating PSD via FFT analysis over the time range 2008-2019 in two land-free areas of the Mediterranean Basin: i) one area across the Central/Aegean Basin ; ii) one area across the Algerian Basin/Sardinian Channel, both depicted in (Fig. 15-c), similarly to the analyses presented for the OSSE.

Both regions are known as dynamically active areas in the Mediterranean Basin (Pujol and Larnicol (2005)). In particular, the KE spectra were inter-compared against the theoretical predictions of two-dimensional turbulence (Vallis (2006)), i.e., the $k^{-3}$ and $k^{-5/3}$ slopes.

In the Central/Aegean area (Fig. 15-a), the spectral analysis confirms the improvement brought by the CNN reconstruction. The SR and Altimeters KE spectra are super-imposed for small wavenumbers, indicating a similar description of the large mesoscale motions and are aligned with the predictions of energy/enstrophy transfer, following the $k^{-3}$ slope for larger wavenumbers and the $k^{-5/3}$ slope for smaller ones.

The improvement of our methodology with respect to standard altimetry processing is evidenced by overall higher PSDs at larger wavenumbers and by a closer alignment with the $k^{-3}$ slope for wavenumbers $\geq 4$ degrees$^{-1}$, i.e. scales $\leq 30$ km (although not fully recovered through the entire range). This reflects a more efficient representation of the small mesoscale features due to our reconstruction. For the Algerian/Sardinian area, the analyses are sketched in the Fig. 15-b and led to similar conclusions as for the Central/Aegean area, although both KE estimates (Altimeter and SR) show less agreement with the $k^{-5/3}$ at wavenumbers $\leq 3$ degrees$^{-1}$. For both areas the SR KE spectra drops at scales around 30 km (Wavenumbers $\simeq 3$ deg$^{-1}$), which can be considered as the scale at which mesoscale features are fully characterized.

## 4   Discussion and Conclusions

In recent years, CNNs for super resolution provided benefits for oceanographic applications, exploiting single and multi-variate approaches (Ducournau and Fablet (2016); Lima et al. (2017); Buongiorno Nardelli et al. (2022); Wang and Li (2023); Fanelli et al. (2024a)). Here, we present a CNN based, multivariate approach to jointly reconstruct ocean surface currents and SST from EU Copernicus satellite data. We trained a CNN model from numerical model outputs and then applied the CNN model to satellite derived data collected over the Mediterranean Area.

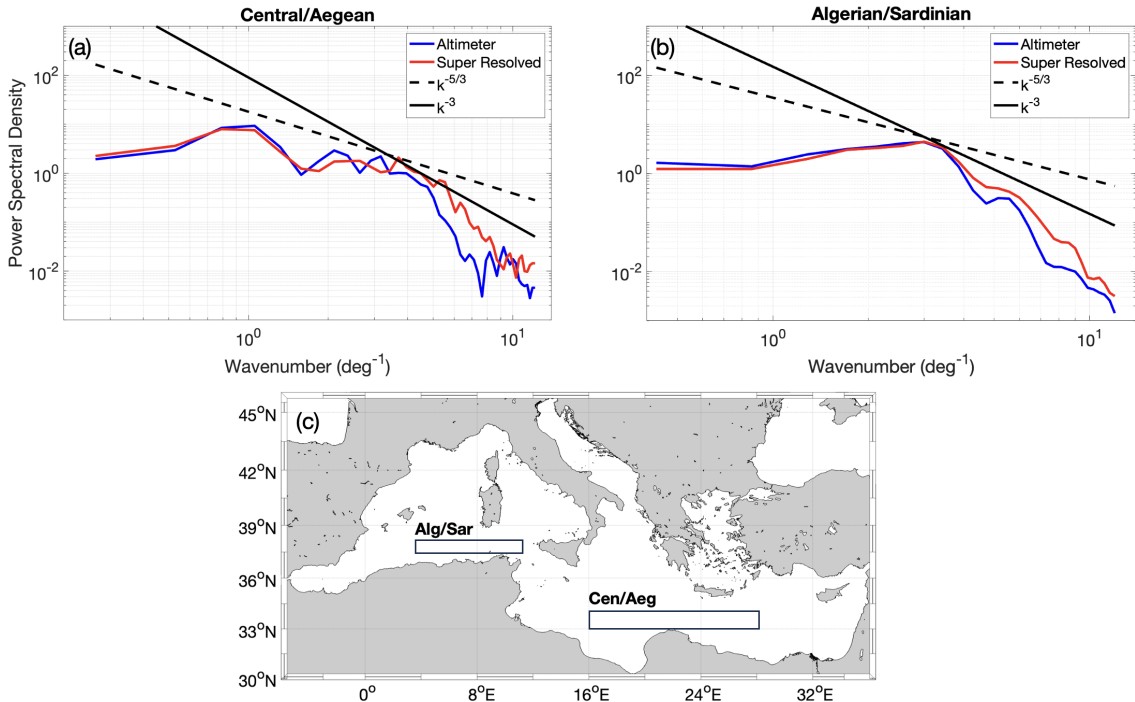

**Figure 15.** Comparative spectral analyses of the KE maps derived from standard Altimeter (blue) and SR (red) ADTs. Panels (a) and (b) respectively refer to the Central/Aegean (Cen/Aeg) and Algerian/Sardinian (Alg/Sar) areas, depicted in panel (c).

With respect to previous similar exercises, we relied here on a new OSSE approach/CNN training architecture, also simulating and integrating issues/artifacts found in the present-day L4 SST satellite products (e.g. Ciani et al. (2020)), as pointed out in Section 2. As such, this study provides a more realistic implementation of the CNN for estimating satellite derived surface currents from ADT and SST and includes the possibility to simultaneously predict ADT and SST fields (although we mostly focused on the reconstruction of ADTs and geostrophic currents).

The OSSE study also addressed potential overfitting issues of the CNN training strategy previously adopted by BBN22. Here, the neural network validation is performed systematically excluding tiles observed in a typical late autumn/early winter period, so that CNN prediction capabilities are tested in a worst case scenario. In other words, the CNN is pushed to predict ocean circulation features in periods of enhanced small mesoscale/submesoscale activity (e.g. Callies et al. (2015)) never seen during training. The OSSE results indicate that the CNN improves the characterization of ocean circulation patterns, intensities and spectral properties even for periods excluded during the training phase, although showing weaker performance with respect to the rest of the year (Figs. 8, 9). This is likely justifying the overall reduced improvement of the OSSE results compared to the BBN22 formulation (Fig. 7). However, it should also be kept in mind that the degraded synthetic SE SST employed in our study is surely impacting these statistics as well, while previous tests were based on an excessively optimistic set-up.

The application of the dADR-SR model to satellite-derived data is assessed by deriving the ocean surface currents from the CNN derived ADTs, relying on the geostrophic approximation. Such currents are then inter-compared with standard altimeter derived currents using: i) in-situ measurements as a benchmark and relying on RMS errors as validation metrics; ii) spectral analysis, to quantify the gain in the description of the mesoscale features. The CNN derived currents presented here, compared to standard altimetry products and to previous findings of BBN22, are able to reduce the RMS error by approximately $1 \ \mathrm{cm \ s^{-1}}$ for the meridional flow and $0.5 \ \mathrm{cm \ s^{-1}}$ for the zonal one. This is likely due to the adoption of an upgraded CNN, which is now trained at OSSE level considering realistic observing systems characteristics for both ADT and SST and includes a physics informed constraint on the LF. Degradations of the CNN derived currents with respect to the Altimetry products mostly occur in near-shore areas, potentially highlighting the limitation of employing L4 SSTs to super-resolve the surface coastal circulation. In coastal areas it is generally harder to interpret or model land contamination effects or to capture small scale, fast coastal processes, thus affecting the quality of optimally interpolated SST products. Also, the physics informed constraint on the LF may fail in coastal environments, where small-scale SST features do not necessarily evolve due to geostrophic advection but can result from ageostrophic processes as e.g. coastal upwelling (particularly pronounced in the Gulf of Lion, Sicily Strait and south of Crete). The characterization of the mesoscale features from standard altimetry and SR ADTs was made via spectral analysis. It confirmed an enhancement of the mesoscale activity obtained via the neural network approach. However, the representation of the 2D geostrophic motion was not fully achieved. The KE spectra obtained from the SR ADTs depicted by Fig. 15 do not fully recover the theoretical energy KE spectrum at all scales. The use of an optimally interpolated SST is certainly a major limit for our reconstruction methodology. OI SSTs are indeed known to smooth/distort some oceanic features as a drawback of the interpolation algorithm (as pointed out recently by González-Haro et al. (2024)). In addition, under prolonged cloud cover, (although the neural network is fed with the information on the SST mapping error) the extraction of features becomes questionable, as the SST fields tend to a daily climatology.

Future studies on the application presented here could include: i) modifying the training strategy for the CNN structure, in which the OSSE could consider ADT/ADT derived quantities and SST as predictors and surface currents (instead of High Resolution ADTs) as targets. This would have the main advantage to go even beyond the prediction of purely geostrophic motions, as achieved in the present study, where surface currents are derived from the geostrophic approximation equations; ii) testing a new prediction of super-resolved surface currents from satellite derived data (as presented in Section 3.2), upon provision of improved L4 SSTs over long timeseries (i.e. at least decadal). Examples of improvements for SST L4 satellite products were recently presented by Sunder et al. (2020); Jung et al. (2022); Fanelli et al. (2024a) and mainly propose an enhancement of feature resolution on the reconstructed L4 fields. This is thus expected to improve the description of dynamical features in both SST and, as in the present study, in L4 fields enriched with dynamical information extracted from the gap-free SST maps. The dynamical coherence of such fields could also be quantified by employing recently published metrics based on multi-fractal theory of turbulence (González-Haro et al. (2024)); iii) reproducing the OSSE discussed in Section 2.6 over time-series longer than one year. Exploiting one full year for training, validation and test would make our results more robust and discard any CNN overfitting issue. Recent works on the SSH mapping based on neural networks pointed out that training based on increasingly larger time series significantly improve the network prediction performances (Archambault et al. (2024)).

Finally, our approach could be adapted to directly learn an end to end mapping between present-day operational ADTs obtained from constellations of nadir looking altimeters and higher-resolution observations from the SWOT wide swath instrument. Operational L4 ADTs are presently optimally interpolated from along-track observations and achieve effective resolutions $O(100 \text{ km})$ at mid latitudes (Ballarotta et al. (2019a); Pujol et al. (2016)). With SWOT, SSH can be observed with resolutions down to 20 km (Fu and Ubelmann (2014); Le Guillou et al. (2021)), with the additional advantage of being native 2D over a $\simeq 120$ km wide swath. As such, potential misrepresentations of the upper ocean dynamics by the numerical models used for the OSSE would be bypassed by directly training on high resolution observations. In all cases, a successful implementation of dADR-SR would allow to carry out a full reprocessing of standard ADT time-series back to the start of satellite radar altimetry era and a potential application to near real time processing.

*Data availability.* The data used for the CNN training, as well as the super-resolved geostrophic currents are available through https://zenodo.org/records/10727432. The satellite derived gridded, gap free ADTs and SSTs are freely available upon registration via https://marine.copernicus.eu/

## Appendix A: Customization of the Loss Function for Super-Resolution

### A1 Determination of the Loss Function hyper-parameters

We show the rationale behind the choice of the hyper-parameters for the LF appearing in Eq. (2). This is shown here for the first two terms of Eq. 2, leading to the determination of the $\alpha$ and $\beta$ parameters. Fig. A1 shows the behavior of the train/validation loss curves for the CNN employing the Loss function given in Eq. (2) considering only the ADT or SST term, separately. This is shown for the first three epochs, as those curves converge quite quickly afterwards.

The ratio between the ADT and SST train/validation loss curves can be quantified by comparing the curves and, from epoch 3 onward is around 4. As such, considering we impose $\alpha$=1, $\beta$ is set to 0.25 in order to equally weight ADT and SST contributions. Such an exercise has been repeated for all the other terms of the loss function, leading to the hyper-parameters appearing in the LF.

### A2 Tuning of the Physics-Informed term

The Physics Informed term $\text{Loss}_{\text{Phy}}$ in Equation (2) was shaped according to the following findings. In a first attempt, such term was expressed by Equation (A1),

$$\text{Loss}_{\text{Phy}} = \left( \frac{\partial \text{SST}}{\partial \text{t}} \right)_{\text{pred}} - \frac{\text{g}}{\text{f}} \frac{\partial \text{ADT}_{\text{pred}}}{\partial \text{y}} \frac{\partial \text{SST}_{\text{pred}}}{\partial \text{x}} + \frac{\text{g}}{\text{f}} \frac{\partial \text{ADT}_{\text{pred}}}{\partial \text{x}} \frac{\partial \text{SST}_{\text{pred}}}{\partial \text{y}} \tag{A1}$$

The reader is referred to Section 3.1 for further details on the Equation. Minimizing such $\text{Loss}_{\text{Phy}}$ term equals asking the predicted SST and ADT to obey to the horizontal geostrophic SST advection. During the determination of the LF hyper-

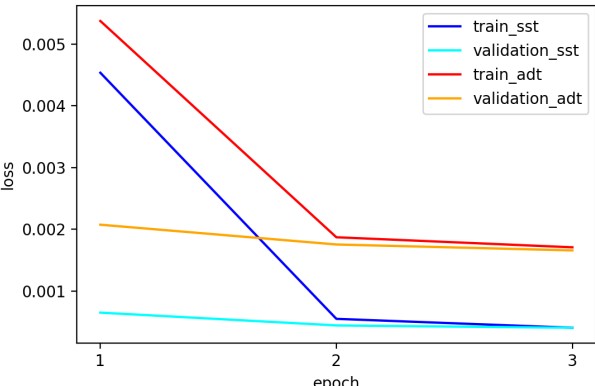

**Figure A1.** Train/validation losses employing a loss function built from ADT (red, orange, respectively) and SST (blue,cyan, respectively) separately.

parameters, (as explained in Section A1) we obtained $\delta = 1.38$. After training the neural network and reconstructing the ocean currents with satellite derived data, the comparison to in-situ measured currents yielded the RMS errors detailed in Table A1 ("Linear" $\text{Loss}_{\text{Phy}}$ Type).

Interestingly, in order to get a further RMS error reduction, we had to decrease the $\delta$ factor manually, which made the approach rather empirical. We however noticed that the $\text{Loss}_{\text{Phy}}$ term given by Eq. (A1) was linear, unlike the ADT, SST and $\partial$tSST terms appearing in Equation (2) , for which the minimization is given in a least squared sense. We thus decided to homogenize all the terms contributing to the Loss function, modifying $\text{Loss}_{\text{Phy}}$ as follows:

$$\text{Loss}_{\text{Phy}} = \overline{\left( \left( \frac{\partial \text{SST}}{\partial \text{t}} \right)_{\text{pred}} - \frac{\text{g}}{\text{f}} \frac{\partial \text{ADT}_{\text{pred}}}{\partial \text{y}} \frac{\partial \text{SST}_{\text{pred}}}{\partial \text{x}} + \frac{\text{g}}{\text{f}} \frac{\partial \text{ADT}_{\text{pred}}}{\partial \text{x}} \frac{\partial \text{SST}_{\text{pred}}}{\partial \text{y}} \right)^2} \tag{A2}$$

In this way, the weighting factor $\delta = 1.38$ automatically adjusted to $\delta = 0.025$ and the effect on the reconstructed surface currents was an improvement for both components of the surface motion (i.e. a lower RMS error compared to in-situ measured currents), as expressed by Table A1. We thus decided to adopt the quadratic formulation for the $\text{Loss}_{\text{Phy}}$ term of the LF.

**Table A1.** RMS error computed by means of in-situ measured currents for the two formulations of the $\text{Loss}_{\text{Phy}}$ term. U and V stand for zonal and meridional currents, respectively.

| $\text{Loss}_{\text{Phy}}$ **Type** | RMS U $(\text{cms}^{-1})$ | RMS V $(\text{cms}^{-1})$ |
|:---:|:---:|:---:|
| **Linear (Eq. A1)** | 12.18 | 12.09 |
| **Quadratic (Eq. A2)** | 11.90 | 11.85 |

*Author contributions.* D.C., C.F. and B.B.N participated to the writing of original manuscript. D.C. performed the CNN training and ocean currents reconstruction/validation. B.B.N. and C.F. contributed to shape the CNN architecture. B.B.N. and D.C. were responsible of the funding acquisition.

*Competing interests.* The Authors declare that no competing interests are present

*Acknowledgements.* The Authors thank the two anonymous Reviewers for providing constructive comments on the manuscript. This study was funded by the European Space Agency (ESA) as part of the World Ocean Circulation (WOC) project, ESA contract no. 4000130730/20/I-NB. Some of the inputs of this study were produced as part of the ESA ocean CIRculation from ocean COLour observations (CIRCOL) project (ESA contract no. 4000128147/19/I-DT).

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
