# Peer review of "Estimating ocean currents from the joint reconstruction of absolute dynamic topography and sea surface temperature through deep learning algorithms"

_EGUsphere, 2024_

## Referee Comment (RC1)

**Estimating ocean currents from the joint reconstruction of absolute dynamic topography and sea surface temperature through deep learning algorithms**

Daniele Ciani , Claudia Fanelli , and Bruno Buongiorno Nardelli

This work present an Observing System Simulation Experiment to improve the spatial resolution of Absolute Dynamic Topography, as a primary objective and Sea Surface Temperature as a secondary objective, based in deep learning methodologies. As a first step, the authors used model outputs to synthesize satellite observations and train a Convolutional Neural Network model, that is later applied using real satellite observations to retrieved 12 years (2008-2019) of higher spatial resolution ADT and geostrophic currents. The study is focused in the Mediterranean Sea, where without any doubt this approach can be challenging and have big impact since the Rossby deformation radius is small (10 km). The manuscript represent a substantial contribution to improve spatial resolution of surface ocean currents retrieved from satellite observations. I think  the manuscript can be slightly  improved before it can be accepted; therefore, I recommend minor revisions. I detail my major concerns bellow.

**Major comments.**

- I found that the manuscript describes in detail the deep learning methodologies used, the improvements made with respect to previous works, and the validation of the Convolutional Neural Network model. However, I think section 3.2 where the trained neural network is used to predict super resolved ADT from satellite altimetry and SST is unbalanced. I found it way too short, and in my opinion, it is one of the substantial contributions of this work. The authors have reconstruct super resolved ADT and derived geostrophic currents for the period 2008-2019. They validate the resulting current fields with in situ currents measured by drifters, providing Root Mean Square error as a metric. The RMS provides information about the accuracy, i.e, it provides an estimation of how well the model is able to predict the target value. It is indeed a good metric, however I would suggest the authors to consider other metrics that can assess the dynamical quality of the retrieved fields. Is this approach valid anytime of the year, or on the contrary it has similar limitations as the ones they stated in the introduction reported by previous works (González-Haro and Isern-Fontanet, 2014; Rio and Santoleri, 2018; Ciani et al., 2020). I am aware extending way far the validation in section 3.2 can be even out of the scope of the manuscript, but I think this deserve at least further attention and discussion.

- The proposed CNN approach enhances the characterization of mesoscale dynamics of current altimetry observations, it is undeniable with the spectral analysis shown in Fig. 4 and 5. However, I find misleading the following affirmations, although they are right:
  - *l 233 : Progressively approaching smaller scales, i.e. from $\simeq$100 km downward (1 deg−1 wavenumber onward), the Super Resolved ADT spectrum (SR-ADT, red line in Fig. 4 (c)) evolves in fair good agreement with the ground-truth (green line in Fig. 4 (c)), confirming an improved representation of smaller mesoscale features compared to standard altimetry products.*
  - *L 236 The SR-ADT spectrum eventually shows the injection of noise below scales of $\simeq$ 20 km, as confirmed by a flattening of the spectrum.*

  Although I do agree with the former affirmations, I think the authors should be more clear and state that the effective spatial resolution of the super resolved ADT is about 50 km  (2 deg^{-1}). This is wavelength in which the PSD deviates from the ground truth (green curve). It is to say, from 100 km to 20 km the PSD of the super resolved ADT is closer to the ground-truth,

when compared to the satellite PSD, but it has already lost energy. I would also suggest the authors to include the theoretical spectral slope curve k^{-5/3} in Fig 4c and Fig5c, for completeness and to facilitate interpreting the PSD curves.

- As briefly introduce in a point earlier, I would suggest the authors to further discuss about the fact that even the effective spatial resolution is improved, the description of dynamical features at the surface may be not guaranteed (l385)

**Minor comments:**

- I would suggest the authors to rephrase the abstract and state that the primary objective is to improve the spatial resolution of ADT.
- Line 52 Consider also other references here: González-Haro et al 2020, Miracca-Lage et al. 2022
- Line 56 I think it could be convenient to state here that the avarage revisit time of  SWOT is about 11 days. It provides higher spatial resolution but the temporal one is much limitated.
- Line 194: *In particular, we forced the validation dataset to be a time series of samples adjacent in time (during the late fall/early winter season), instead of applying a random selection from the available samples.* Justify here why, it is stated further in the text line 357: *In other words, the CNN is pushed to predict ocean circulation features in periods of enhanced small mesoscale/submesoscale activity (e.g. Callies et al. (2015)) never seen during training.*
- Line *197 four predictors: namely the SE-ADT, SE-ADT error, SST and its temporal derivatives ($\partial tSST$ )* shouldn't "SST" here also be SE-SST?
- Figure 7: suggestion: could you mark in different colors the dates corresponding to cases shown in  Fig. 4, 5 and 8?
- In general, and because there are a number of datasets it is difficult to follow the resulting spatial resolution of retrieved fields. I am assuming all of them are giving at the same spatial resolution than the model: 1/24 degrees. It can be deduced from the Power spectral analysis. I am also assuming that the retrieved fields from satellite observations  in section 3.2 is 1/24, please state it clearer in the text, even in the abstract.

---

## Author Comment (AC1)

**Reply to Reviewer 1**

**Estimating ocean currents from the joint reconstruction of absolute dynamic topography and sea surface temperature through deep learning algorithms**

By Daniele Ciani , Claudia Fanelli , and Bruno Buongiorno Nardelli

This work present an Observing System Simulation Experiment to improve the spatial resolution of Absolute Dynamic Topography, as a primary objective and Sea Surface Temperature as a secondary objective, based in deep learning methodologies. As a first step, the authors used model outputs to synthesize satellite observations and train a Convolutional Neural Network model, that is later applied using real satellite observations to retrieved 12 years (2008-2019) of higher spatial resolution ADT and geostrophic currents. The study is focused in the Mediterranean Sea, where without any doubt this approach can be challenging and have big impact since the Rossby deformation radius is small (10 km). The manuscript represent a substantial contribution to improve spatial resolution of surface ocean currents retrieved from satellite observations. I think the manuscript can be slightly improved before it can be accepted; therefore, I recommend minor revisions. I detail my major concerns bellow.

We thank the Reviewer for the constructive comments provided. Please find below our replies to the specific points.

Major comments.
- I found that the manuscript describes in detail the deep learning methodologies used, the improvements made with respect to previous works, and the validation of the Convolutional Neural Network model. However, I think section 3.2 where the trained neural network is used to predict super resolved ADT from satellite altimetry and SST is unbalanced. I found it way too short, and in my opinion, it is one of the substantial contributions of this work. The authors have reconstruct super resolved ADT and derived geostrophic currents for the period 2008-2019. They validate the resulting current fields with in situ currents measured by drifters, providing Root Mean Square error as a metric. The RMS provides information about the accuracy, i.e, it provides an estimation of how well the model is able to predict the target value. It is indeed a good metric, however I would suggest the authors to consider other metrics that can assess the dynamical quality of the retrieved fields. Is this approach valid anytime of the year, or on the contrary it has similar limitations as the ones they stated in the introduction reported by previous works (González-Haro and Isern-Fontanet, 2014; Rio and Santoleri, 2018; Ciani et al., 2020). I am aware extending way far the validation in section 3.2 can be even out of the scope of the manuscript, but I think this deserve at least further attention and discussion.

We thank Reviewer 1 for providing this comment on the manuscript. In order to balance the results between the numerical experiment and the analyses based on satellite data, we added spectral analyses to inter-compare the power spectral density (PSD) of the surface kinetic energies (KE) derived from standard Altimeter (up-sized to the 1/24° grid) and super-resolved ADTs, to have insights on the effective spatial resolution of the two datasets. This analysis was performed using a fast Fourier transform (FFT) over the time range 2008-2019 in two land-free areas of the Mediterranean Basin: i) one area across the Central/Aegean Basin (33.7°N-34.5°N--16.5°%-27.7°E) ; ii) one area across the Algerian Basin/Sardinian Channel (37.4°N-38.3°N--4°E-11.4°E), both depicted in Figure 1.R1-c of the present document. Both regions are known as dynamically active areas in the Mediterranean Basin (Pujol and Larnicol, 2005). In particular, the KE spectra were inter-compared against the theoretical predictions of two-dimensional turbulence (Vallis, 2006)), i.e., the $k^{-3}$ and $k^{-5/3}$ slopes. Especially in the Central/Aegean area (Figure 1.R1-a), the spectral analysis confirms the improvement brought by the CNN reconstruction.

The SR and Altimeters KE spectra are super-imposed for small wavenumbers, indicating a similar description of the large mesoscale motions and are aligned with the predictions of energy/enstrophy transfer, following the $k^{-3}$ slope for larger wavenumbers and the $k^{-5/3}$ slope for smaller ones. The improvement of our methodology with respect to standard altimetry processing is evidenced by overall higher PSDs at larger wavenumbers and by a closer alignment with the $k^{-3}$ slope for wavenumbers $\geq 4$ degrees$^{-1}$, i.e. scales $\leq 30$ km (although not fully recovered through the entire range). This reflects a more efficient representation of mesoscale features associated with our reconstruction. For the Algerian/Sardinian area, the analyses are sketched in the Figure 1.R1-b below and led to similar conclusions as for the Central/Aegean area, although both KE estimates (Altimeter and SR) show less agreement with the $k^{-5/3}$ at wavenumbers $\leq 3$ degrees$^{-1}$.

[Figure]

Figure 1.R1: Comparative spectral analyses of the KE maps derived from standard Altimeter (blue) and SR (red) ADTs. Panels (a) and (b) respectively refer to the Central/Aegean and Algerian/Sardinian areas, depicted in panel (c).

Such results are now introduced in the revised version of the manuscript . Additionally, in the discussion section, we introduced a hint for future validation metrics to quantify feature resolution in the reconstructed L4 Fields (lines 373-391, lines 440-443 of the revised manuscript).

Moreover, we tried to assess the performances of the CNN-based ADT (and derived surface currents) reconstruction as a function of the season. The winter (December to February) and summer (June to August) validation of the super-resolved geostrophic currents with respect to in-situ measurements are reported in Figure 2.R1. In both seasons we can recognize an overall improvement of the Super-Resolved (SR) surface currents with respect to standard Altimetry. In winter (Figure 3.R1-a) the RMS error reduction of the SR currents is around 1 cm/s for both components of the surface flow; in Summer (Figure 2.R1-b) we have comparable performances for the zonal flow and a 0.5 cm/s RMS error reduction of the SR currents, for the meridional flow. It is however really challenging to assess the seasonal behaviour of such statistics, as the numbers of in-situ measurements is not comparable across seasons (see e.g. Fig 2.R1-c,d). In the manuscript, we would thus just keep the overall statistics in order to rely on a larger number of satellite/in-situ matchups throughout the basin.

[Figure]

Figure 2.R1 **a)** Differences of RMS errors between the Altimeter derived (Alti) and SR currents during WINTER: (top) zonal flow, (bottom) meridional flow. Red areas express an improvement with respect to standard altimetry. **b)** Differences of RMS errors between the Altimeter derived (Alti) and SR currents during SUMMER: (top) zonal flow, (bottom) meridional flow. Red areas express an improvement with respect to standard altimetry. **c), d)** number of in-situ measurements during winter and summer, respectively. Statistics are provided in 2°x2° boxes.

- The proposed CNN approach enhances the characterization of mesoscale dynamics of current altimetry observations, it is undeniable with the spectral analysis shown in Fig. 4 and 5. However, I find misleading the following affirmations, although they are right:

  - l 233 : Progressively approaching smaller scales, i.e. from 100 km downward (1 deg−1 ≃100 km downward (1 deg−1 wavenumber onward), the Super Resolved ADT spectrum (SR-ADT, red line in Fig. 4 (c)) evolves in fair good agreement with the ground-truth (green line in Fig. 4 (c)), confirming an improved representation of smaller mesoscale features compared to standard altimetry products.
  - L 236 The SR-ADT spectrum eventually shows the injection of noise below scales of 20 ≃100 km downward (1 deg−1 km, as confirmed by a flattening of the spectrum.

  Although I do agree with the former affirmations, I think the authors should be more clear and state that the effective spatial resolution of the super resolved ADT is about 50 km (2 deg^{- 1}). This is wavelength in which the PSD deviates from the ground truth (green curve). It is to say, from 100 km to 20 km the PSD of the super resolved ADT is closer to the ground-truth, when compared to the satellite PSD, but it has already lost energy. I would also suggest the authors to include the theoretical spectral slope curve k^{-5/3} in Fig 4c and Fig5c, for completeness and to facilitate interpreting the PSD curves.

  Thanks for providing these comments. In order to evaluate the relative behaviour of our outputs with respect to the $k^{-5/3}$ spectral slope, we inter-compared the Kinetic Energy (KE) spectra of the dataset involved in our study (as in Ciani et al. 2019 and following Vallis 2017). The KE are computed using the surface currents derived from the Satellite equivalent (SE) , Super Resolved (SR) ADTs and from the model outputs (GROUND TRUTH).

[Figure]

[Figure]

Figure 3.R1. Comparative spectral analysis of the Kinetic Energy maps. Results refer to the 2D box depicted in the top panel . The black continuum and dashed lines represent the predictions of the 2D Energy transfer (k^-5/3) and 2D enstrophy cascade (k^-3), respectively.

As expected, the modelled surface currents (our reference) follow the prediction of the 2D turbulence energy/enstrophy cascade, in agreement with Vallis 2017. In particular, for smaller wavenumbers, the KE spectrum (green line) is closer to the $k^{-5/3}$ slope, while it mostly follows the $k^{-3}$ for larger ones. An expected exception occurs for wavenumbers approaching 10 deg$^{-1}$, where the spectrum exhibits an energy loss, suggesting that model outputs are unable to fully resolve submesoscale motion. Interestingly, the KE spectrum obtained from SR-ADT currents (in red) is pretty much in line with the evolution of our reference case. A slight energy reduction is observed in the mesoscale range, but the improvement with respect to standard synthetic altimetry is evident. We thus decided to add this figure and comment it in the revised manuscript, also adding information on the fully resolved scales (Please see lines 259-271 of the revised manuscript).

- As briefly introduce in a point earlier, I would suggest the authors to further discuss about the fact that even the effective spatial resolution is improved, the description of dynamical features at the surface may be not guaranteed (l385)

We thank the Reviewer for this comment. We agree on the fact that an improved effective spatial resolution (confirmed by the spectral analyses for both the OSSE and the application to satellite-derived data) does not guarantee the full description of small mesoscale features at the ocean surface with the dADR-SR methodology. For instance, one can see that the KE spectra obtained from the Super-Resolved ADT depicted by figures 1.R1 and 3.R1 of the

present document do not fully recover the theoretical energy KE spectrum at all scales. The use of an optimally interpolated SST is certainly a major limit for our reconstruction methodology. OI SSTs are indeed known to smooth/distort some oceanic features as a drawback of the interpolation algorithm. In addition, under prolonged cloud cover, (although we are here feeding the network with this information) the extraction of features becomes questionable, as the SST fields tend to reproduce the smoother background field. This could be overcome in future studies relying on a training based on L3C SSTs (as done for single-image super-resolution by Fanelli et al. 2024), which has been already identified as one future development of the present study (See lines 425-432 of the revised manuscript). Following the Reviewer's suggestion, we inserted an additional comment on this topic at lines 440-443 of the discussion section, also introducing one comment on feature resolution evaluation with additional metrics.

**Minor comments:**

- I would suggest the authors to rephrase the abstract and state that the primary objective is to improve the spatial resolution of ADT.

  This has been corrected in the manuscript, please see lines 1-3 of the revised version.

- Line 52 Consider also other references here: González-Haro et al 2020, Miracca-Lage et al. 2022

  The references have been added, as suggested. Please double check line 56 of the revised manuscript.

- Line 56 I think it could be convenient to state here that the avarage revisit time of SWOT is about 11 days. It provides higher spatial resolution but the temporal one is much limitated.

  Thanks for this comment, which has now been integrated in the revised manuscript. Please check lines 63-64 of the revised manuscript.

- Line 194: In particular, we forced the validation dataset to be a time series of samples adjacent in time (during the late fall/early winter season), instead of applying a random selection from the available samples. Justify here why, it is stated further in the text line 357: In other words, the CNN is pushed to predict ocean circulation features in periods of enhanced small mesoscale/submesoscale activity (e.g. Callies et al., 2015) never seen during training.

  Thanks for pointing this out. We introduced a brief comment on this at lines 201-204 of the revised manuscript, enabling the readers to have an initial idea of the reason behind our choice and referring to Section 4 for more details.

- Line 197 four predictors: namely the SE-ADT, SE-ADT error, SST and its temporal derivatives ($\partial_t$SST ) shouldn't "SST" here also be SE-SST?

  Thanks for this comment. It actually made us realize it was worth further specifying this point.  At line 197 of the original manuscript,  the SST is indicated as is because it refers to the former CNN architecture by Buongiorno Nardelli et al. 2022, in which the SST was simply extracted from the model outputs and did not account for the satellite processing effects, it was thus assumed as perfectly known. The SST is thus a SE-SST only in the present study. In order to further clarify this, we modified the sentence from
  *"In previous formulations, the network considered"* to
  *"In previous formulations (Buongiorno Nardelli et al. 2022)"*, the network considered
  Please, check line 206 of the revised manuscript.

- Figure 7: suggestion: could you mark in different colors the dates corresponding to cases shown in Fig. 4, 5 and 8?

  Thanks for this comment. The image (now Figure 8 in the revised manuscript) , as well as its caption, have been modified accordingly

- In general, and because there are a number of datasets it is difficult to follow the resulting spatial resolution of retrieved fields. I am assuming all of them are giving at the same spatial resolution than the model: 1/24 degrees. It can be deduced from the Power spectral analysis. I am also assuming that the retrieved fields from satellite observations in section 3.2 is 1/24, please state it clearer in the text, even in the abstract.

  We thank Reviewer 1 for this suggestion. A specific comment has been inserted in the abstract and along the manuscript, at lines 8,14-15, 110-111, 121-122, 150-152, 168-170, 179-180  respectively.

**References**

Buongiorno Nardelli, B., Tronconi, C., Pisano, A., and Santoleri, R.: High and Ultra-High resolution processing of satellite Sea Surface

Temperature data over Southern European Seas in the framework of MyOcean project, Remote Sensing of Environment, 129, 1–16, 2013.

Buongiorno Nardelli, B., Cavaliere, D., Charles, E., and Ciani, D.: Super-resolving ocean dynamics from space with computer vision algorithms, Remote Sensing, 14, 1159, 2022.

Ciani, D., Rio, M. H., Menna, M., & Santoleri, R. (2019). A synergetic approach for the space-based sea surface currents retrieval in the Mediterranean Sea. Remote Sensing, 11(11), 1285.

Fanelli, C., Ciani, D., Pisano, A., & Buongiorno Nardelli, B. (2024). Deep learning for the super resolution of Mediterranean sea surface temperature fields. Ocean Science, 20(4), 1035-1050.

Pujol, M. I., & Larnicol, G. (2005). Mediterranean sea eddy kinetic energy variability from 11 years of altimetric data. Journal of Marine Systems, 58(3-4), 121-142.

Vallis, G. K. (2017). Atmospheric and oceanic fluid dynamics. Cambridge University Press.

---

## Author Comment (AC2)

**Reply to Reviewer 2**
**Estimating ocean currents from the joint reconstruction of absolute dynamic topography and sea surface temperature through deep learning algorithms**

By Daniele Ciani , Claudia Fanelli , and Bruno Buongiorno Nardelli

The authors are following their previous work on super-resolving and inpainting sea-surface height, this time focusing on a network learned through a new OSSE experiment. Their new study focuses on the Mediterranean Sea and is used to evaluate sea currents.
There are significant strong points in their approach. The article is well structured and scientifically sound, and can further a very active research field tied to the OceanChallenges data challenge. I especially appreciate the care taken to evaluate the physical fields obtained.

We thank the Reviewer for the constructive comments provided . Please find below our replies to the specific points.

In general, the authors are keenly aware of the bibliography in the field. However, given the many similitudes (there are differences too) I would like to see their positioning in regards to Archambault, Théo, et al. "Learning sea surface height interpolation from multi-variate simulated satellite observations." Journal of Advances in Modeling Earth Systems 16.6 (2024): e2023MS004047.

We thank the reviewer for suggesting this reference, which was added in the introduction and discussion section (please double check lines 68, 443-446 of the revised manuscript). There are indeed similarities between these two approaches, as both exploit the "transfer learning problem" from numerically simulated data to real-world data for the mapping of sea surface height (SSH), exploiting the combined use of SSH and Sea Surface Temperature (SST) data. Additionally, both methodologies result from fine-tuning/comparing different types of Loss function to train the neural network. Reading the manuscript we noticed the major following advantages (+) and disadvantages (-) of their approach, compared to ours.

Archambault et al. 2024 (A24 hereinafter)

+ Unlike our study, A24 rely on a significantly much longer time series to train the neural network, as they use a 20 years time series of simulated SSH/SST data, also including along-track altimeter observations, and they rely on one full year for the test. Such an approach certainly reduces the possibility of residual overfitting issues, as pointed out in our study.

+ The Availability of one full year for the final test is indeed extremely interesting, enabling one to assess the reconstruction performances as a function of the season.
+ The A24 set-up allows both supervised and unsupervised training, making the reconstruction methodology more applicable directly to satellite-derived observations
- According to the OSSE results, the A24 methodology is preferably applicable to predict observations in a temporal window centered at +/- 10 days, indicating that the methodology is more fit for delayed-time mode application. In an operational context, conversely, the methodology presented in our manuscript (e.g. using NRT SSH/SST L4 maps or L4 SSH and L3 SST maps) would result immediately applicable for near-real-time (NRT) processing.
- To the best of our understanding, the generation of the simulated satellite observations adopted in Archambault et al., although providing a realistic satellite-equivalent L3 dataset for SSH and SST, does not reproduce the optimal interpolation processing used within actual operational services (as done in our study for both SST and SSH data). We believe that our approach (in which we mimic both the SSH and SST 2D mapping artifacts introduced by the current algorithms within the Copernicus operational production) could be more directly implemented to improve present-day operational SSH and SST. We already identified the possibility of extending our OSSE in time and to account for training relying on "real-world observations". We thus believe it is worth mentioning this by making a link with the A24 publication. We inserted a comment on this at lines 443-446 of the revised manuscript.

I have some reservations in regards to the validation procedure since it has some contradictory information in the paragraph that starts at line 183. There seems to be attention paid to avoid data leakage but at the same time, early in the paragraph, the 40 days seem to be selected randomly. A clarification of which is the actual approach in this paper, and how it guarantees a significant enough time lag between data used in train, validation, and test is important.

In this work, the 40 dates kept aside for the validation are not purely random selected samples. As stated at line 192 of the revised manuscript, those dates are selected on purpose in order to perform the final test over different dynamical regimes, i.e. accounting for the full seasonality. Figure 8 indeed shows that the Julian Days over which we performed the final test are fairly equally distributed throughout the four seasons. Such 40 dates only represent ~11% of the entire time series. The rest of the time series is only used for training/validation purposes. The novelty of this work, which is summarized by figure 3 and explained at lines 200-204, 402-406 of the revised manuscript, lies in a different training/validation strategy. In particular, training is performed systematically excluding samples at the end of our time series i.e. the ones characterized by late autumn/winter dynamics. In other words, during training, the neural networks never "sees" features of the typical late autumn/winter period, where a more frequent presence of small scale features is expected (in full agreement with the findings of Callies et al. 2015). In this way, the neural network is asked to infer beyond what explicitly

learned in previous studies (e.g. BBN22). In BBN22, the validation procedure was based on randomly selected samples (from the ~89% percent of the timeseries used for training/validation purposes). In this way, it is likely that the neural network was trained/validated using samples that are too close to each other in time, potentially generating overfitting issues. In the newly proposed approach, the occurrence of overfitting can be reasonably excluded, as discussed at lines 320-327 of the revised manuscript. Moreover, the new architecture demonstrated to outperform the reconstruction illustrated by BBN22, confirming the validity of our upgrades.

However, we agree with the Reviewer that our approach still presents some limits. Relying on a single year for training/validation and test is suboptimal. Future applications plan to build an OSSE based at least on one full year for each of the following operations: training, validation and test, as claimed in the manuscript at lines 327-329, 443-446 of the revised manuscript.

The choice of architecture is interesting, and I expect that a lot of other approaches were tested. It would be interesting to include them in the annexes, as negative results are often ill-represented in literature.

Thanks for this comment. As pointed out at lines 78-80 of the revised manuscript, several architectures have been tested and thoroughly discussed in a former study, ranging from the baseline CNN configuration proposed by Dong et al. 2015 to the dADR-SR developed by Buongiorno Nardelli et al. 2022 (BBN22 hereinafter). Here, as mentioned at lines 81-85 of the revised manuscript, we investigated the effect of introducing simple, yet substantial improvements to the dADR-SR, which outperformed the other configurations tested by BBN22. The dADR-SR , in the former implementation, considered a non-realistic input SST field, limiting the potential of accounting for more sophisticated loss functions. In our study we thus introduced realistic SST / SST error fields which allowed us to modify:

1) The set of input/output data for the CNN training (as described at lines 210-212 of the revised manuscript)
2) The Loss function, with the introduction of a Physics-Informed approach (lines 288-313 of the revised manuscript)

Nevertheless, our achievements on the dADR-SR with the physics informed approach was not straightforward and required a couple of tests in order to understand how to shape the new, customized loss function given by eq (2) of the revised manuscript.

In a first attempt, the Physics Informed term of the Loss Function (LF) was given by (1.R2) and its Loss_phy term was expressed by (2.R2)

$$\text{LF} = \alpha\overline{[(\text{SST}_{\text{pred}} - \text{SST}_{\text{ref}})^2]} + \beta\overline{[(\text{ADT}_{\text{pred}} - \text{ADT}_{\text{ref}})^2]} + \gamma\overline{\{[(\partial_t\text{SST})_{\text{pred}} - (\partial_t\text{SST})_{\text{ref}}]^2\}} + \delta\text{Loss}_{\text{phy}} \quad (1.\text{R2})$$

$$\text{Loss}_{\text{Phy}} = \left(\frac{\partial\text{SST}}{\partial t}\right)_{\text{pred}} - \frac{g}{f}\frac{\partial\text{ADT}_{\text{pred}}}{\partial y}\frac{\partial\text{SST}_{\text{pred}}}{\partial x} + \frac{g}{f}\frac{\partial\text{ADT}_{\text{pred}}}{\partial x}\frac{\partial\text{SST}_{\text{pred}}}{\partial y} \quad (2.\text{R2})$$

Minimizing the Loss_phy term as in (2.R2) equals asking the predicted SST and ADT to obey to the horizontal geostrophic SST advection. When performing the initial test to find the weights of the Loss_phy term "delta" (as explained at lines 460-468 of the revised manuscript) we obtained delta=1.38. After training the neural network and reconstructing the ocean currents with satellite derived data, the comparison to in-situ measured currents yielded the results detailed in table t1.R2. Interestingly, in order to get a further RMS error reduction, we had to reduce the "delta" factor manually, which made the approach rather empirical.

We however noticed that the Loss_phy term given by (2.R2) was linear, unlike the ADT, SST and ∂tSST terms appearing in 1.R2, for which the minimization is given in a least squared sense. We thus decided to homogenize all the terms contributing to the Loss function , modifying Loss_phy as in (3.R2)

$$\text{Loss}_{\text{Phy}} = \overline{\left(\left(\frac{\partial\text{SST}}{\partial t}\right)_{\text{pred}} + \frac{g}{f}\frac{\partial\text{ADT}_{\text{pred}}}{\partial y}\frac{\partial\text{SST}_{\text{pred}}}{\partial x} + \frac{g}{f}\frac{\partial\text{ADT}_{\text{pred}}}{\partial x}\frac{\partial\text{SST}_{\text{pred}}}{\partial y}\right)^2} \quad (3.\text{R2})$$

In this way, the weighting factor "delta" automatically adjusted to 0.025 and the effect on the reconstructed surface currents was an improvement for both components of the surface motion (i.e. a lower Root Mean Square error compared to in-situ measured currents), as expressed by table t1.R2. We thus decided to adopt the quadratic formulation for the Loss_phy term (shown in (3.R2)).

| | RMS zonal (cm/s) | RMS meridional (cm/s) |
|---|---|---|
| Loss_phy "linear" (eq. 2.R2) | 12.18 | 12.09 |

| Loss_phy "quadratic" (eq. 3.R2) | 11.90 | 11.85 |
|---|---|---|

Table t1.R2. Root mean square error computed by means of in-situ measured currents for both components of the surface currents and for the two formulations of the Loss_phy term.

We added such comments in Appendix A1, please see lines 470-485 of the revised manuscript

There is also a technical question of how the authors reconstruct the whole Mediterranean basin, and whether there are discontinuities in the full reconstruction.

The reconstruction of the Mediterranean Basin ADT (and derived geostrophic currents) is operated tile by tile. However, such tiles are not simply arranged adjacent to each other, as they consider a 50% overlap along the zonal and meridional directions (longitude-latitude). When reconstructing the tiles, we expect enhanced performances in the central area of the tile and potential spurious features towards the tile periphery, due to edge effects related to convolutional kernels. To overcome this, the reconstruction of the basin-scale field is obtained accounting for the longitude-latitude overlap but also applying a pixel wise weighting function that assigns a progressively decreasing weight to pixels lying at larger distances from the tiles centers. Such a methodology enables a seamless basin-scale reconstruction starting from the 76x100 tiles. We inserted a comment on that at lines 230-236 of the revised manuscript.

More details would be useful as to the process of selecting the hyperparameters of the new loss function, and some ablations on its usefulness would not be remiss.

We provide as an example the way we tuned the hyperparameters (alpha, beta) for the first two terms of the loss function, that we report below for convenience (equation 4.R2)

$$LF = \alpha[\overline{(SST_{pred} - SST_{ref})^2}] + \beta[\overline{(ADT_{pred} - ADT_{ref})^2}] \quad \text{(4.R2)}$$

[Figure]

Figure 1.R2. Train validation losses employing a loss function built from ADT (red, orange, respectively) and SST (blue,cyan, respectively) separately.

The figure shows the behavior of the train/validation loss curves for the CNN employing the Loss function given by Eq. 4.R2, considering only the ADT or SST term, separately. This is shown for the first three epochs, as those curves converge quite quickly afterwards. The ratio between the ADT and SST train/validation loss curves can be quantified by comparing the curves and, from epoch 3 onward is around 4. As such, considering we impose alpha=1, beta has to be 0.25 in order to equally weight ADT and SST contributions. Such an exercise has been repeated for all the other terms of the loss function reported in the manuscript and has led to the final loss functions with the hyperparameters described at lines 460-468 of the revised manuscript. This was inserted in Appendix A1, together with the tuning of the physics - informed term. Please go to lines 470-485 of the revised manuscript.

While the evaluation of the currents is extremely important, the method relies on geostrophic approximation, which holds in the Mediterranean, but should be discussed in the case of applying this approach to other basins closer to the equator where this approximation does not hold.

We thank the Reviewer for this comment. The dADR-SR, in our approach, was built to improve the satellite derived ADT (as also stressed by Reviewer 1) and derived geostrophic circulation in the Mediterranean Sea. We believe this approach could be easily adaptable to other mid-latitude areas of the ocean. Equatorial areas could be studied via simple updates of the present-day architecture in which the training is not based on ADT but directly on surface currents. For example, the training of the CNN from numerical model outputs and based on the Observing System Simulation Experiment approach would account for the following datasets:
   1) The satellite equivalent SST, $\partial t$SST and their error fields - (INPUT);

2) The satellite-equivalent-derived geostrophic currents (including as well the error field) - (INPUT)
3) The model-derived SST and $\partial t$SST - (TARGET)
4) The model-derived total surface currents - (TARGET)

In this way the neural network would be optimized to reconstruct directly the two components of the surface currents. This approach is also part of future studies on this topic, as already mentioned in the discussion section of the manuscript (see lines 433-436 of the revised manuscript)

In general, should the authors address these small points I would be very pleased to see their work published.

**References**

Buongiorno Nardelli, B., Cavaliere, D., Charles, E., and Ciani, D.: Super-resolving ocean dynamics from space with computer vision algorithms, Remote Sensing, 14, 1159, 2022

Callies, J., Ferrari, R., Klymak, J. M., and Gula, J.: Seasonality in submesoscale turbulence, Nature Communications, 6, 6862, 2015.

Dong, C., Loy, C. C., He, K., and Tang, X.: Image super-resolution using deep convolutional networks, IEEE transactions on pattern analysis and machine intelligence, 38, 295–307, 2015

---

## Author Response (AR2)

**Reply 2 to Reviewer 1**

**Estimating ocean currents from the joint reconstruction of absolute dynamic topography and sea surface temperature through deep learning algorithms**

By Daniele Ciani , Claudia Fanelli , and Bruno Buongiorno Nardelli

The authors have addressed my concerns in the first round of the manuscript. I found the manuscript has improved and it is ready for publication after a few minor typos corrections or suggestions.

We thank the Reviewer for the constructive comments on our manuscript and for providing additional help to identify typos and inconsistencies. Please find below a point-by-point reply.

Minor typos, number of line referred to the track changes manuscript:

l.75 typo "resoultion"
Thanks. This has been corrected, please check line 73 of the newly revised manuscript.

l. 123 cut off wavelength
We considered adding this expression, please check line 120-121 of the newly revised manuscript.

Caption Fig 6. Remove parenthesis (Kinetic Energy spectra of
Thanks. This has been corrected, please check Figure 6 of the newly revised manuscript.

l. 273 until scales of ≃ 40 km? I think is clear in 50km
Thanks. We modified it to 50 km, please check line 270 of the newly revised manuscript.

l .375 inter-compare inter compare (be cohesive)
Thanks for finding this inconsistency, we adopted the "inter-compare" version through the entire manuscript.

l. 431 Consider to include a reference here: as shown recently by González-Haro et al. 2024
We considered adding this statement and related reference. Please check line 430 of the newly revised manuscript.

On behalf of the Authors
Daniele Ciani